# Leveraging augmented-Lagrangian techniques for differentiating over infeasible quadratic programs in machine learning

**Antoine Bambade**
Inria - Département d'Informatique de l'École normale supérieure, PSL Research University,
École des Ponts, Marne-la-Vallée, France.
`{bambade.antoine}@gmail.com`

**Fabian Schramm, Adrien Taylor, Justin Carpentier**
Inria - Département d'Informatique de l'École normale supérieure, PSL Research University.
`{fabian.schramm,adrien.taylor,justin.carpentier}@inria.fr`

## Abstract

Optimization layers within neural network architectures have become increasingly popular for their ability to solve a wide range of machine learning tasks and to model domain-specific knowledge. However, designing optimization layers requires careful consideration as the underlying optimization problems might be infeasible during training. Motivated by applications in learning, control and robotics, this work focuses on convex quadratic programming (QP) layers. The specific structure of this type of optimization layer can be efficiently exploited for faster computations while still allowing rich modeling capabilities. We leverage primal-dual augmented Lagrangian techniques for computing derivatives of both feasible and infeasible QP solutions. More precisely, we propose a unified approach that tackles the differentiability of the closest feasible QP solutions in a classical $\ell_2$ sense. We then harness this approach to enrich the expressive capabilities of existing QP layers. More precisely, we show how differentiating through infeasible QPs during training enables to drive towards feasibility at test time a new range of QP layers. These layers notably demonstrate superior predictive performance in some conventional learning tasks. Additionally, we present alternative formulations that enhance numerical robustness, speed, and accuracy for training such layers. Along with these contributions, we provide an open-source C++ software package called QPLayer for differentiating feasible and infeasible convex QPs and which can be interfaced with modern learning frameworks.

## 1 Introduction

Incorporating differentiable optimization problems as layers within neural networks has recently become practical and effective for solving certain machine learning tasks, see, for instance (Geng et al., 2020; Amos & Kolter, 2017; Lee et al., 2019; Le Lidec et al., 2021; Donti et al., 2017; de Avila Belbute-Peres et al., 2018; Amos et al., 2018; Bounou et al., 2021; Donti et al., 2021b;a). Such layers allow capturing useful domain-specific knowledge or priors. Unlike conventional neural networks, where the output of each layer is provided by a simple (explicit) function of its input, the input of an optimization layer is the parameter of an optimization problem, and its output is a solution to this problem. Figure 1 and Figure 2 provide two illustrative examples of a neural network and a QP layer. Both layers have potentially fixed (in blue) and trained (in red) parameters. The main difference is that the output (i.e., $y^\star$ in Figure 1) of the feed-forward neural network has a closed-form expression, whereas the output of the QP layer is the solution of a constrained QP (i.e., $y^\star$ in Figure 2). Finally, note that extra parameters (i.e., $z^t$ and $h^t$ in red in Figure 2) are trained to ensure the quadratic program is always well-posed during training.

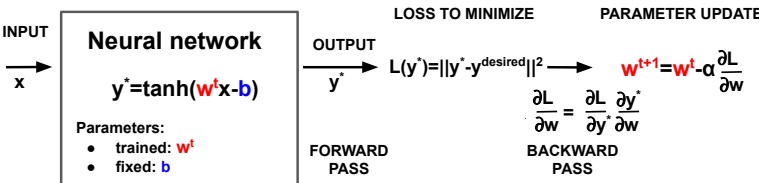

Figure 1: Example of a feed-forward neural network.

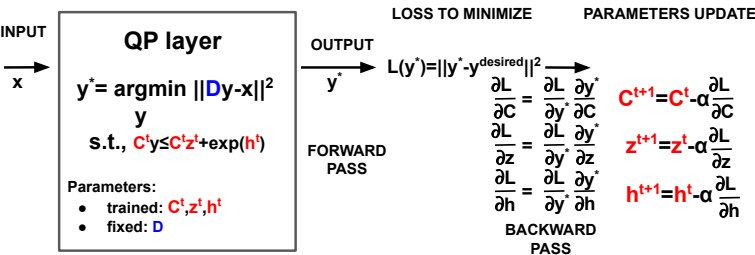

Figure 2: Example of a Quadratic Programming layer (with $D$ nonsingular).

In this work, we focus on convex Quadratic Programming (QP) layers, a specific type of optimization layer that offers a rich modeling power (Amos & Kolter, 2017, Section 3.2). A convex QP parameterized by $\theta$ is defined as follows

$$x^{\star}(\theta) \in \operatorname*{argmin}_{x \in \mathbb{R}^n} \left\{ f(x;\,\theta) := \frac{1}{2} x^{\top} H(\theta)\, x + x^{\top} g(\theta) \right\} \tag{QP($\theta$)}$$
$$\text{s.t. } C(\theta)\, x \leqslant u(\theta),$$

where $H(\theta) \in \mathcal{S}_{+}^{n}(\mathbb{R})$ is a real symmetric positive semi-definite matrix of $\mathbb{R}^{n \times n}$, $g(\theta) \in \mathbb{R}^n$, $C(\theta) \in \mathbb{R}^{n_i \times n}$ and $u(\theta) \in \mathbb{R}^{n_i}$. $n$ is the problem dimension, while $n_i$ is the number of inequality constraints. We will abusively denote $H$, $g$, $C$, and $u$ without explicit dependence on $\theta$ when this dependence is clear from the context or does not generate any ambiguity. In order to use QP($\theta$) as a learning tool that can be trained with standard optimization techniques, we need to be able to differentiate $x^{\star}(\theta)$ w.r.t. $\theta$, which is challenging for a few reasons. First, there is usually no practical way to compute a closed-form for $x^{\star}(\theta)$, even when QP($\theta$) is well-defined. Second, even when such an $x^{\star}(\theta)$ exists, there is no guarantee for it to be unique nor differentiable w.r.t. $\theta$ (see, e.g., the assumptions of the implicit function theorem (Dontchev & Rockafellar, 2009, Theorem 1B.1)). As a consequence, concurrent approaches are generally based on architectures enforcing satisfaction of some strong assumptions. In particular, to the best of our knowledge, previous approaches specialized for differentiating through QP($\theta$) enforce primal feasibility of the layer during training, which generally requires additional learning variables and limits the modeling power of those layers. For instance, as in (Amos & Kolter, 2017), learning QP($\theta$) requires imposing its feasible set to be non-empty. For imposing this while learning $C$, the authors also learn $z \in \mathbb{R}^n$ and $h \in \mathbb{R}^{n_i}$ and $u$ of the form $u = Cz + \exp(h)$ (similarly to Figure 2), thereby preventing, among others, $u$ from being fixed independently of the learning.

This work makes the following contributions:

- We propose a unified approach to tackle the differentiability of both feasible and infeasible QPs. The main idea consists in extending the definition of $x^{\star}(\theta)$ to be either a solution to QP($\theta$) when it is feasible or a solution of the closest feasible QP (in the least-square sense) when it is not. By relying on the notion of conservative Jacobian by (Bolte et al., 2021; Bolte & Pauwels, 2020), we notably show that the KKT map $G$ of this extended problem is path differentiable w.r.t. $\theta$ and $x^{\star}$ (Section 3.2). In this context, the Jacobian $\frac{\partial x^{\star}(\theta)}{\partial \theta}$ is defined as the least-square solution of the linear system formed by applying the implicit function theorem to $G$ Section 3.3. We show that this definition consistently covers the differentiability of feasible QPs as with the traditional implicit differentiation (Amos & Kolter, 2017) when it is valid and with the least-square estimate proposed by (Agrawal et al., 2019, Appendix B) otherwise.

- In Section 3.4 we provide efficient ways to compute the Jacobian $\frac{\partial x^{\star}(\theta)}{\partial \theta}$ in forward and backward automatic differentiation modes.

- In Section 4 we demonstrate how the approach enables dealing with possibly infeasible QP($\theta$) during training, while converging for test time to a feasible layer. We illustrate how it allows to train a broader range of QP layers (e.g., learning QPs that are not generically feasible). More precisely, we will show how to drive towards feasibility at test time the QP layer provided in Figure 3. Learning $A^t$ (in red) is not obvious since nothing guarantees a priori that the fixed equality constraint vector (of ones) lies in the range space of $A^t$. We will see that learning such layer notably provides better predictive power for some classic learning tasks.

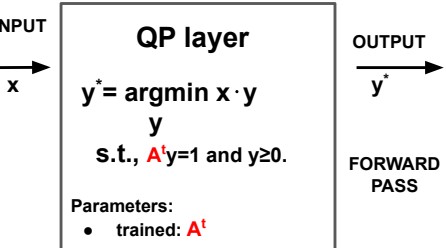

Figure 3: A Linear Programming layer. Nothing guarantees during training that the constrained vector of 1 lies in the range space of the trained matrix $A^t$. Our approach enables to train $A^t$ such that at test time the LP is feasible.

Based on these developments, we provide QPLayer, an open-source implementation with efficient forward and backward passes, freely available at `https://github.com/Simple-Robotics/proxsuite`. It takes advantage, among others, of recent advances in solving of QP problems to output in the forward pass the closest feasible QP solution in $\ell_2$-sense as soon as the program is primal infeasible (Chiche & Gilbert, 2016). Section 4 highlights for different learning tasks the numerical robustness, accuracy, and speed of our approach against other state-of-the-art methods. Moreover, Appendix C.1 provides different simple experiments with parametric QPs to illustrate the solutions provided by QPLayer in different cases (e.g., LP case, primal infeasible QP case etc.). Appendix C.2 contains additional timing experiments. Appendix C.3 illustrates through several experiments that QPLayer is numerically more robust and can thereby be trained with large learning rates.

## 2  RELATED WORK

**Differentiation for optimization layers.** Under certain regularity conditions, it is possible to implicitly differentiate the optimality conditions of convex optimization problems. (Gould et al., 2016; Gilbert, 2021; Fiacco & McCormick, 1968; Robinson, 1980) present general conditions and techniques under which it is possible to differentiate through constrained optimization problems using the implicit function theorem. (Bolte et al., 2021; Bolte & Pauwels, 2020) present extensions to nonsmooth (not necessarily differentiable) functions for machine learning and optimization applications. (Amos & Kolter, 2017) treats the specific case of QPs with a dedicated network architecture, OptNet, and a specialized batched interior-point solver, Qpth, which allows for efficient backpropagation. More recently, (Butler & Kwon, 2023) proposed an alternative approach based on the differentiation of ADMM steps, yet dealing only with equality and box inequality constraints[1]. For more general convex optimization, (Amos et al., 2018) proposed CvxpyLayer that differentiates through "Disciplined Convex Programs" using an LSQR (Paige & Saunders, 1982) solver to speed up the differentiation procedure. (Sun et al., 2022) has recently proposed an ADMM-type method, called Alt-Diff, to alternatively solving a constrained convex optimization program and obtaining approximate Jacobians at the current approximate solutions.(Blondel et al., 2022) also proposed a generic solver, JaxOpt, based on an implicit automatic differentiation mechanism leveraging the Jax framework. Finally, let us mention the work (Sharma et al., 2022), which provided a Julia library,

---

[1]The associated R solver is not yet publicly available.

DiffOpt.jl, for differentiable QP and conic optimization (or any model that can be reformulated into these standard forms).

**Unrolling methods.** Argmin and Argmax operations can be approximated by first-order methods, which can be unrolled (Domke, 2012; Monga et al., 2021). These architectures typically introduce an optimization procedure such as gradient descent into the inference procedure (Belanger & McCallum, 2016; Belanger et al., 2017; Amos et al., 2017; Metz et al., 2019), which is usually truncated to a predefined number of iterations. Recently, (Scieur et al., 2022) highlighted the "curse of unrolling" by showing that for unconstrained quadratic optimization, there is a tradeoff between the convergence speed of the iterates and that of the Jacobian. Although unrolling methods are easy to implement, most of their applications are limited to unconstrained problems. Indeed, if constraints are added, the unrolling solutions have to be projected into the feasible region, significantly increasing the computational burdens.

**Implicit models.** Implicit models replace explicit expressions in neural networks with layers defined by implicit functions (Zhang et al., 2020). As for optimization layers, the backward pass requires solving nonlinear Jacobian-based equations arising from the implicit function theorem. Recently, there have been a growing number of applications using them, such as neural ODE (Chen et al., 2018), deep equilibrium models (Bai et al., 2019), logical reasoning in deep neural network (using MAXSAT SDP relaxation) (Wang et al., 2019), implicit surface representation (Michalkiewicz et al., 2019), attention mechanisms (Geng et al., 2021a), graph neural networks (Gu et al., 2020). However, this method is not suitable for optimization layers with complicated constraints. Recently, (Fung et al., 2022) proposes a matrix-free approach to decrease computational costs, and (Geng et al., 2021b) proposes a phantom gradient that relies on fixed-point unrolling and a Neumann series for faster computations of approximate update directions.

## 3 THE EXTENDED CONSERVATIVE JACOBIAN FOR CONVEX QPS

This section introduces the main contribution of this work: an extended conservative Jacobian for the solutions to QP($\theta$) allowing to simultaneously deal with feasible and infeasible QPs, as provided in Section 3.2 and Section 3.3. Section 3.4 proposes efficient algorithms for computing them in forward and backward modes. For exposition purposes, Appendix C.1 illustrates the concepts on a few simple examples.

### 3.1 PROBLEM FORMULATION

For differentiating QPs, we solve a hierarchic problem QP-H($\theta$) which is equivalent to QP($\theta$) when QP($\theta$) is primal feasible (i.e., there exists $x$ s.t. $C(\theta)x \leqslant u(\theta)$)

$$s^\star(\theta) = \arg\min_{s \in \mathbb{R}^{n_i}} \tfrac{1}{2}\|s\|_2^2$$
$$\text{s.t. } x^\star(\theta), z^\star(\theta) \in \arg\min_{x \in \mathbb{R}^n} \max_{z \in \mathbb{R}^{n_i}_+} L(x, z, s; \theta), \quad \text{(QP-H($\theta$))}$$

with $L(x, z, s; \theta) := \tfrac{1}{2}x^\top H(\theta)x + x^\top g(\theta) + z^\top(C(\theta)x - u(\theta) - s)$ (namely the Lagrangian of QP($\theta$) augmented with a slack variable $s$). The following assumption is necessary and sufficient for guaranteeing QP-H($\theta$) to have a solution. In this situation, QP-H($\theta$) is therefore well-posed and $s^\star$ is referred to as the optimal shift. It provides a measure of the distance of QP($\theta$) to be primal infeasible in $\ell_2$-sense (hence $s^\star = 0$ iff QP($\theta$) is feasible).

**Assumption 1.** *$H(\theta)$ is symmetric positive definite in the direction of $g(\theta)$ or $g(\theta)$ is orthogonal to the recession cone of QP($\theta$), i.e., $g(\theta) \perp \mathcal{C}^\infty(\theta) := \{y \in \mathbb{R}^n | C(\theta)[x + \tau y] \leqslant u(\theta) \text{ s.t. } C(\theta)x \leqslant u(\theta), \tau \geqslant 0\}$.*

The existence of a solution $(x^\star(\theta), z^\star(\theta), s^\star(\theta))$ is also equivalent to the dual of QP($\theta$) having a non-empty domain (i.e., being proper), see (Chiche & Gilbert, 2016, Assumption 2.6 and Proposition 2.5)). So, the approach proposed here allows differentiating through dual feasible convex QPs.

## 3.2 THE CLOSEST FEASIBLE QP

In what follows, we deal with QP-H($\theta$) via a nonlinear map $G$:

$$G(x, z, t; \theta) := \begin{bmatrix} H(\theta)x + g(\theta) + C(\theta)^\top z \\ C(\theta)x - u(\theta) - t \\ [[t]_- + z]_+ - z \\ C(\theta)^\top [t]_+ \end{bmatrix}, \tag{G}$$

where $[.]_+$ and $[.]_-$ respectively correspond to component-wise projections on the non-negative and non-positive orthants. The following lemma guarantees solutions to QP-H($\theta$) to be zeros of $G$ (see proof in Appendix A).

**Remark 1.** *The map $G$ is found via a change of variable of the KKT conditions for QP-H($\theta$).*

**Lemma 1.** *Let $H(\theta) \in \mathcal{S}_+^n(\mathbb{R})$, $g(\theta) \in \mathbb{R}^n$, $C(\theta) \in \mathbb{R}^{n_i \times n}$ and $u(\theta) \in \mathbb{R}^{n_i}$ be satisfying Assumption 1. It holds that $(x^\star, z^\star, s^\star)$ solves QP-H($\theta$) iff there exists $t^\star \in \mathbb{R}^{n_i}$ s.t. $G(x^\star, z^\star, t^\star; \theta) = 0$ and $s^\star = [t^\star]_+$.*

## 3.3 THE EXTENDED CONSERVATIVE JACOBIAN

For differentiating through $G$, we rely on the notion of extended conservative Jacobian (ECJ). As provided by the following lemma, the nonlinear map $G(x, z, t; \theta)$ is path differentiable (see (Bolte & Pauwels, 2020, Definition 3)) w.r.t. $x$, $z$, $t$ and also w.r.t. $\theta$ under the assumption that $H(\theta)$, $g(\theta)$, $C(\theta)$ and $u(\theta)$ are differentiable w.r.t. $\theta$. This lemma is proved in Appendix B.

**Lemma 2.** *$G$ is path differentiable w.r.t. $x^\star$, $z^\star$ and $t^\star$. Furthermore, if $H(\theta)$, $g(\theta)$, $C(\theta)$ and $u(\theta)$ are differentiable w.r.t. $\theta$, then $G$ is path differentiable w.r.t. $\theta$.*

**Definition 1.** *Let $H(\theta)$, $g(\theta)$, $C(\theta)$ and $u(\theta)$ be differentiable w.r.t. $\theta$ and satisfying Assumption 1. Let $v^\star = (x^\star, z^\star, t^\star) \in \mathbb{R}^n \times \mathbb{R}_+^{n_i} \times \mathbb{R}^{n_i}$ s.t. $G(x^\star, z^\star, t^\star; \theta) = 0$. We refer to the ECJs of $x^\star$, $z^\star$ and $t^\star$, respectively denoted by $\frac{\partial x^\star}{\partial \theta}, \frac{\partial z^\star}{\partial \theta}$ and $\frac{\partial t^\star}{\partial \theta}$, as solutions of the following problem*

$$\left( \frac{\partial x^\star}{\partial \theta}, \frac{\partial z^\star}{\partial \theta}, \frac{\partial t^\star}{\partial \theta} \right) \in \arg\min_w \left\| \frac{\partial G(x^\star, z^\star, t^\star; \theta)}{\partial v^\star} w + \frac{\partial G(x^\star, z^\star, t^\star; \theta)}{\partial \theta} \right\|_2^2. \tag{1}$$

*Furthermore, we refer to an ECJ of $s^\star = [t^\star]_+$, denoted by $\frac{\partial s^\star}{\partial \theta}$, any element satisfying $\Pi \frac{\partial t^\star}{\partial \theta} \in \frac{\partial s^\star}{\partial \theta}$, with $\Pi \in \partial([.]_+)(t^\star)$ a subgradient of the positive orthant evaluated in $t^\star$.*

As shown in the next section the ECJs match the definitions of standard Jacobians under standard assumptions guaranteeing differentiability (when the QP is feasible), as provided by (Amos & Kolter, 2017; Dontchev & Rockafellar, 2009). When the QP is feasible but not differentiable, the ECJ corresponds to a least-square approximation specialized for QPs. A similar practical least-square estimate was proposed in (Agrawal et al., 2019, Appendix B) for differentiating primal solutions of second-order cones (SOCs)[2]. (Blondel et al., 2022, Section 2.1) proposed a similar estimate.

**Remark 2.** *As introduced in Bolte & Pauwels (2020), conservative Jacobians are generalized forms of Jacobians well suited for automatic differentiation. A locally Lipschitz function is called path differentiable if it has a conservative Jacobian. Importantly, path differentiability is equivalent to having a chain rule for the Clarke subdifferential.*

## 3.4 DERIVING AN EXTENDED CONSERVATIVE JACOBIAN

This section derives an ECJ and incorporates it in a backpropagation algorithm. It also shows how to efficiently compute this ECJ under primal feasibility.

### 3.4.1 GENERAL CASE: DEALING WITH BOTH FEASIBLE AND INFEASIBLE QPS

In the following, we provide forward and backward pass algorithms to compute ECJs for both feasible and infeasible QPs.

---

[2]More precisely, (Agrawal et al., 2019, Appendix B) relies on a series of assumptions allowing to simplify the computations. In particular, they assume that $\frac{\partial z^\star}{\partial \theta} = 0$ and $\frac{\partial t^\star}{\partial \theta} = 0$, where $t^\star$ is a slack variable.

**Forward pass:** Let $H(\theta)$, $g(\theta)$, $C(\theta)$ and $u(\theta)$ be differentiable w.r.t. $\theta$ and satisfying Assumption 1, and $x^\star, z^\star, t^\star$ s.t. $G(x^\star, z^\star, t^\star; \theta) = 0$. We show in Appendix B.2.1.1 that we can efficiently derive ECJs of $x^\star$, $z^\star$ and $t^\star$ by solving the following QP using an augmented Lagrangian-based algorithm (Rockafellar, 1976)

$$\frac{\partial x^\star}{\partial \theta}, \frac{\partial z^\star}{\partial \theta}, \frac{\partial t^\star}{\partial \theta} \in \underset{\frac{\partial x}{\partial \theta}, \frac{\partial z}{\partial \theta}, \frac{\partial t}{\partial \theta}}{\arg\min} \left\| \begin{bmatrix} H & C^\top & 0 \\ C & 0 & -I \\ 0 & \Pi_1 - I & \Pi_1 \Pi_2 \\ 0 & 0 & C^\top(I - \Pi_2) \end{bmatrix} \begin{bmatrix} \frac{\partial x}{\partial \theta} \\ \frac{\partial z}{\partial \theta} \\ \frac{\partial t}{\partial \theta} \end{bmatrix} + \begin{bmatrix} \frac{\partial H}{\partial \theta}x^\star + \frac{\partial g}{\partial \theta} + \frac{\partial C}{\partial \theta}^\top z^\star \\ \frac{\partial C}{\partial \theta}x^\star - \frac{\partial u}{\partial \theta} \\ 0 \\ \frac{\partial C}{\partial \theta}^\top [t^\star]_+ \end{bmatrix} \right\|_2^2, \tag{2}$$

where $\Pi_1$ and $\Pi_2$ are binary diagonal matrices respectively corresponding to the subdifferentials $\partial([.]_+)([t^\star]_- + z^\star)$ and $\partial([.]_-)(t^\star)$, with the following specific choices in zeros

$$(\Pi_1)_i = 1 \text{ when } [t_i^\star]_- + z_i^\star = 0, \qquad\qquad (\Pi_2)_i = 1 \text{ when } t_i^\star = 0. \tag{3}$$

Furthermore, an ECJ of $s^\star$ can be obtained via $(1 - \Pi_2)\frac{\partial t^\star}{\partial \theta} \in \frac{\partial s^\star}{\partial \theta}$. As $s^\star$ is a direct output of an augmented Lagrangian-based algorithm (Chiche & Gilbert, 2016), in what follows, we work with $s^\star$ instead of $t^\star$.

**Backward pass:** Let $h : \mathbb{R}^n \times (\mathbb{R}^{n_i})^2 \to \mathbb{R}$ be a differentiable function, and let $H(\theta)$, $g(\theta)$, $C(\theta)$ and $u(\theta)$ be differentiable w.r.t. $\theta$ and satisfying Assumption 1 and denote $\mathcal{L}(\theta) := h(x^\star(\theta), z^\star(\theta), s^\star(\theta))$. The following assumption is sufficient for ensuring a conservative Jacobian $\frac{\partial \mathcal{L}}{\partial \theta}$ can be obtained from the usual chain rule:

**Assumption 2.** $x^\star$, $z^\star$ and $t^\star$ are path differentiable w.r.t. $\theta$.

More precisely, under Assumption 2, one can compute:

$$\frac{\partial \mathcal{L}}{\partial \theta} = \frac{\partial \mathcal{L}}{\partial x^\star}^\top \frac{\partial x^\star}{\partial \theta} + \frac{\partial \mathcal{L}}{\partial z^\star}^\top \frac{\partial z^\star}{\partial \theta} + \frac{\partial \mathcal{L}}{\partial s^\star}^\top \frac{\partial s^\star}{\partial \theta}. \tag{4}$$

Following the methodology provided in (Amos & Kolter, 2017, Section 3), we also show in Appendix B.2.1.2 that a conservative Jacobian $\frac{\partial \mathcal{L}}{\partial \theta}$ can be obtained via the following expression, which is computationally more attractive:

$$\frac{\partial \mathcal{L}}{\partial \theta} = (b_1^\star)^\top \frac{\partial H}{\partial \theta}x^\star + (b_1^\star)^\top \frac{\partial g}{\partial \theta} + (b_2^\star)^\top \frac{\partial C}{\partial \theta}x^\star + (z^\star)^\top \frac{\partial C}{\partial \theta}b_1^\star + (s^\star)^\top \frac{\partial C}{\partial \theta}b_4^\star - (b_2^\star)^\top \frac{\partial u}{\partial \theta}, \tag{5}$$

where $b_1^\star, b_2^\star, b_3^\star$ and $b_4^\star$ are solutions of the linear system

$$\begin{bmatrix} H & C^\top & 0 & 0 \\ C & 0 & (I - \Pi_1) & 0 \\ 0 & -I & -\Pi_1\Pi_2 & (1 - \Pi_2)C \end{bmatrix} \begin{bmatrix} b_1^\star \\ b_2^\star \\ b_3^\star \\ b_4^\star \end{bmatrix} = - \begin{bmatrix} \frac{\partial \mathcal{L}}{\partial x^\star} \\ \frac{\partial \mathcal{L}}{\partial z^\star} \\ \frac{\partial \mathcal{L}}{\partial s^\star} \end{bmatrix}. \tag{6}$$

Assumption 3 provides non degeneracy assumptions which ensure equation 6 has solutions:

**Assumption 3.** $C(\theta)$ is full column rank, $s^\star > 0$ (i.e., all constraints are primal infeasible) and $\frac{\partial \mathcal{L}}{\partial z^\star} = 0$ (e.g., the loss $\mathcal{L}$ does not depend of $z^\star$).

### 3.4.2 Exploiting primal feasibility of the QP

In this section, we exploit feasibility of the QP for simplifying the computations. First, for the forward pass, the QP needs only be feasible for the value of $\theta$ under consideration. For the backward pass, we exploit the standard assumption (see (Amos & Kolter, 2017)) of the QP being constructively feasible for all values of $\theta$ (which is of course restrictive, but which can be exploited for efficiency).

**Forward pass:** When QP$(\theta)$ is feasible and $H(\theta)$, $g(\theta)$, $C(\theta)$ and $u(\theta)$ are differentiable w.r.t. $\theta$ and satisfy Assumption 1, we show in Appendix B.2.2.1 that ECJs can be obtained as a solution to the simpler:

$$\frac{\partial x^\star}{\partial \theta}, \frac{\partial z^\star}{\partial \theta} \in \underset{\frac{\partial x}{\partial \theta}, \frac{\partial z}{\partial \theta}}{\arg\min} \left\| \begin{bmatrix} H & C^\top \\ \frac{\Pi_1}{\sqrt{1+\Pi_1}}C & \Pi_1 - I \end{bmatrix} \begin{bmatrix} \frac{\partial x}{\partial \theta} \\ \frac{\partial z}{\partial \theta} \end{bmatrix} + \begin{bmatrix} \frac{\partial H}{\partial \theta}x^\star + \frac{\partial g}{\partial \theta} + \frac{\partial C}{\partial \theta}^\top z^\star \\ \frac{\Pi_1}{\sqrt{1+\Pi_1}}\left(\frac{\partial C}{\partial \theta}x^\star - \frac{\partial u}{\partial \theta}\right) \end{bmatrix} \right\|_2^2, \tag{7}$$

$$\frac{\partial t^\star}{\partial \theta} = (I + \Pi_1)^{-1}\left(C\frac{\partial x^\star}{\partial \theta} + \frac{\partial C}{\partial \theta}x^\star - \frac{\partial u}{\partial \theta}\right), \tag{8}$$

where $\Pi_1$ is a binary diagonal matrices representing the subdifferential $\partial [.]_+ (Cx^\star - u + z^\star)$ with the following specific choice:

$$(\Pi_1)_i = 1 \text{ when } C_i x^\star - u_i + z_i^\star = 0. \tag{9}$$

The following lemma (see proof in Appendix B.2.1.2) guarantees that, under standard assumptions, solutions to equation 7 correspond to standard Jacobians (see, e.g., (Amos & Kolter, 2017)).

**Lemma 3.** *If $QP(\theta)$ is feasible and $H(\theta)$, $g(\theta)$, $C(\theta)$ and $u(\theta)$ are differentiable w.r.t. $\theta$ and satisfy Assumption 1, and if the KKT matrix of active constraints is nonsingular and $x^\star$, $z^\star$ satisfy strict complementarity, then the ECJs matches the standard Jacobian, i.e., $\frac{\partial x^\star(\theta)}{\partial \theta} = \nabla x^\star(\theta)$ and $\frac{\partial z^\star(\theta)}{\partial \theta} = \nabla z^\star(\theta)$.*

**Backward pass:** If $QP(\theta)$ is by construction primal feasible for any $\theta$, then for any $\theta$, $s^\star(\theta) = 0$. We can exploit this result for considering simpler losses not depending anymore of $s^\star(\theta)$. More precisely, let $h : \mathbb{R}^n \times (\mathbb{R}^{n_i}) \to \mathbb{R}$ be a differentiable function, and let $H(\theta)$, $g(\theta)$, $C(\theta)$ and $u(\theta)$ be differentiable w.r.t. $\theta$ and satisfying Assumption 1. Then, denoting $\mathcal{L}(\theta) := h(x^\star(\theta), z^\star(\theta))$, we show in Appendix B.2.2.2 that when assumptions of Lemma 3 hold the backward pass can be evaluated by solving the following linear system

$$\begin{bmatrix} H & C_J^\top \\ C_J & 0 \end{bmatrix} \begin{bmatrix} b_x^\star \\ b_{z_J}^\star \end{bmatrix} = - \begin{bmatrix} \frac{\partial \mathcal{L}}{\partial x^\star} \\ \frac{\partial \mathcal{L}}{\partial z_J^\star} \end{bmatrix}, \qquad b_{z_{J^c}}^\star = \frac{\partial \mathcal{L}}{\partial z_{J^c}^\star}, \tag{10}$$

where $J$ is the set of constraints for which $(\Pi_1)_i = 1$ and $J^c$ the one for which $(\Pi_1)_i = 0$. $\frac{\partial \mathcal{L}}{\partial \theta}$ is then retrieved from the chain rule

$$\frac{\partial \mathcal{L}}{\partial \theta} = (b_x^\star)^\top \frac{\partial H}{\partial \theta} x^\star + (b_x^\star)^\top \frac{\partial g}{\partial \theta} + (\Pi_1 b_z^\star)^\top \frac{\partial C}{\partial \theta} x^\star + (z^\star)^\top \frac{\partial C}{\partial \theta} b_x^\star - (\Pi_1 b_z^\star)^\top \frac{\partial u}{\partial \theta}. \tag{11}$$

Note that the assumptions of Lemma 3 are not necessarily met. Such infeasibility can be detected easily using iterative refinement (Parikh & Boyd, 2014, Section 4.1.2)) as it converges to the least-square solution of equation 10 in the infeasible case (see (Güler, 1991, Theorem 2.3)).

### 3.4.3 FUTURE WORK AND POTENTIAL IMPROVEMENTS

Before moving to the experiments, let us mention a few potential directions for future work and improvements in our approach. There remain a few gaps in the theoretical foundations of our methodology, which we believe should be handled in the future. Indeed, we have not proved that we could apply the chain rule to ECJs in the general case in the spirit of CJs (see (Bolte et al., 2021)). Also, it should be confirmed that ECJ indeed reduces to CJ. From what we can tell, one major unresolved gap is the scope of applicability of the Implicit function Theorem (IFT). Indeed, standard IFT requires at some point nonsingular matrices, which is not compatible with conditions when $QP(\theta)$ is primal infeasible (since primal infeasibility is provoked by degeneracy conditions). Nevertheless, there are known examples (see e.g., the discussion in (Krantz & Parks, 2002, Section 5.4)) when the IFT can still be applied, even when dealing with degenerate matrices. We leave for future work, extension of these techniques for formulating a IFT adapted for our use-case. While those problems are present in most frameworks (Agrawal et al., 2019, Section B), (Blondel et al., 2022, Section 2.1), using the least-square estimate provides good practical results when non-differentiability occurs.

## 4 EXPERIMENTAL RESULTS

Our backward mode differentiation of convex QP layers has been implemented in C++. We refer to it as QPLayer in what follows. Our code leverages the primal-dual augmented Lagrangian solver ProxQP (Bambade et al., 2022), also written in C++ as its internal QP solver. This section illustrates through a classic Sudoku learning tasks that QPLayer allows relaxing primal feasibility constraints, thereby enabling the training of simplified layers.

**Benchmark setup.** QPLayer is compared to OptNet and CvxpyLayer. The experiments were conducted using all threads of an Intel(R) Core(TM) i7-4790 CPU @ 3.60GHz. The benchmark API is available at `https://github.com/Bambade/qplayer_benchmark`.

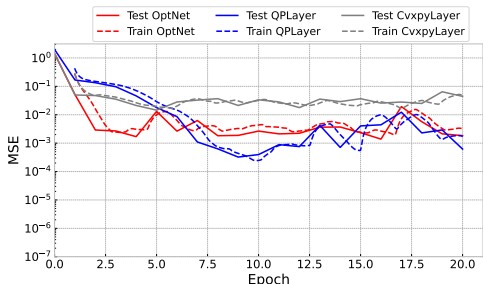 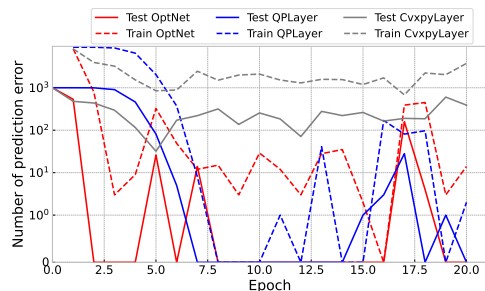

(a) Test and training MSE losses of QPLayer, CvxpyLayer and OptNet layers.

(b) Test and training prediction errors of QPLayer, CvxpyLayer, and OptNet over 1000 and 9000 puzzles.

Figure 4: Sudoku training and test plots using QPLayer, OptNet, and CvxpyLayer. QPLayer can learn LPs (which appear more appropriate), whereas OptNet is restricted to strictly convex QPs.

## 4.1 Learning capabilities

Differentiable optimization for neural network layers has shown great representational power for learning problems that are fundamentally rooted in optimization. The Sudoku problem is one such problem, which can naturally be cast as a mixed integer linear program (MILP). For the Sudoku, OptNet recently showed better robustness and prediction accuracy results than traditional neural networks (Amos & Kolter, 2017, Section 4.4). This section shows that QPLayer generalizes even better by exploiting the fact that it allows learning LPs (and not only QPs). Further, the ability of QPLayer to deal with possibly primal infeasible problems during the learning process appears to be key.

### 4.1.1 Learning linear programs

The Sudoku problem is detailed in (Amos & Kolter, 2017, Section 4.4). We reproduce those experiments with OptNet and CvxpyLayer, while letting QPLayer learn linear programs (LPs) instead of strictly convex QPs (the layer model is detailed in Figure 12 of the appendix). In this experiment, the parameters learned in the layer are the equality constraint matrix and the associated equality constraint vector (through extra variables ensuring the structural feasibility of the layer at training and test time, see Figure 12 of the appendix). OptNet, CvxpyLayer, and QPLayer were trained using Adam with a batch of size 150 and a learning rate of 0.05 to minimize an MSE loss on the dataset created by (Amos & Kolter, 2017). The dataset contains 9000 training puzzles and 1000 held-out puzzles for testing. First, Figure 4a shows that QPLayer manages to reach better local miniminum for the training and test loss (i.e., about an order of magnitude at the 10th iteration). It ends as well over-fitting to the training data, similarly to OptNet and CvxpyLayer. Yet, Figure 4b shows that QPLayer makes at the end of the training process far less prediction errors (i.e., 33 errors for QPLayer over the last 5 epochs, versus 162 for OptNet and 1531 for CvxpyLayer). This advocates that learning a LP instead of a QP enables more accurate and robust training for solving Sudokus.

### 4.1.2 Handling primal infeasibility

As outlined in Section 4.1.1, forcing primal feasibility while learning is a common algorithmic strategy. For the Sudoku problems, those techniques enforcing primal feasibility typically involve neglecting a linear equality constraint $Ax = 1$ (we learn $A$) which corresponds to Sudoku rules. As shown in Figure 5b, this means that the learning procedures do not respect the Sudoku rule constraint (see green and orange dashed lines—labeled "QPLayer; $Ax = 1$ violation" and "OptNet; $Ax = 1$ violation"). In the end, neglecting this constraint ultimately leads to learning a constraint matrix that is inconsistent with the Sudoku rules. Relaxing the primal feasibility imposed by differentiation procedures of previous solvers thereby appears to be key.

By incorporating a potential optimal shift $s^\star$ in its formulation (as exposed in Section 3.1), QPLayer allows dealing with infeasible problems during training. In order to drive towards feasibility the layer at test time (i.e., such that $A$ forms a feasible constraint $Ax = 1$), we consider penalizing the optimal shift $s^\star$ in the learning loss function (see the layer architecture in Figure 13). The backward pass takes then into account the ECJ derivatives introduced in Section 3.4.1. Numerical performances are reported in Figure 5a and Figure 5b. It is apparent that the dark green curve labeled "QPLayer-learn A; $Ax = 1$ violation" converges after the end of the first epoch towards a model satisfying Sudoku

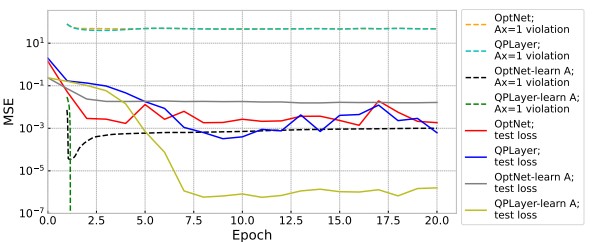 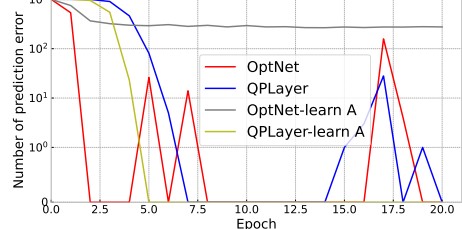

(a) Test MSE loss of QPLayer, OptNet, QPLayer-learn A, and OptNet-learn A specialized for learning A. It includes Sudoku $Ax = 1$ violation.

(b) Test prediction errors over 1000 puzzles of OptNet, QPLayer, QPLayer-learn A and OptNet-learn A specialized for learning A.

Figure 5: Sudoku training and test plots using QPLayer and OptNet layers. QPLayer can learn LPs, whereas OptNet is restricted to strictly convex QPs, which limits its representational power. Contrary to OptNet, QPLayer can be specialized to learn models satisfying specific linear constraints.

rules. The respective yellow prediction and loss curves also converges slightly faster towards a regime without any prediction errors. The steeper slope observed in the graph suggests that it might be worthwhile to train a layer that more accurately adheres to the Sudoku rules, as this could potentially lead to faster puzzle-solving and more interpretable outcomes.

For comparison with OptNet, we have considered reformulating QP-H($\theta$) as a convex QP (see Figure 14 in the appendix). The resulting problem considers more variables and constitutes thus a potentially harder problem to solve. As OptNet can only learn strictly convex QPs, we have also added a small quadratics over the primal variables (similarly to the structural feasible case described in Section 4.1.1). The grey curve "OptNet learn A; test loss" shows the result. It can be seen that it decreases slower and saturates at an earlier level, which is consistent with the fact that the problem is harder to solve. Furthermore, as expected, it displays a worse prediction error. Indeed, the dark dashed curves "OptNet-learn A; $Ax = 1$ violation" outputs the primal feasibility violation. It can be seen that it quickly decreases over the first epochs. Yet, at some point it does not manage to decrease further[3] and saturates at some local minimum around $10^{-3}$.

Finally, let us mention that the formulation developed in Section 3.4 enables a considerable speed-up over the QP reformulation of equation QP-H($\theta$). Indeed, the forward and backward pass using QPLayer account for about $0.45 \pm 0.07$ seconds per batch, whereas OptNet takes over $8.99 \pm 0.91$ seconds per batch.

## 4.2 ADDITIONAL EXPERIMENTS

Appendix C.1 provides different simple experiments with parametric QPs to illustrate ECJs concept. Appendix C.2 contains additional timing experiments. Appendix C.3 illustrates through several experiments that QPLayer is numerically more robust and can thereby be trained with large learning rates.

## 5 CONCLUSION

In this work, we introduced an approach for differentiating both feasible and infeasible convex quadratic programs in a unified fashion. This approach is particularly relevant for learning with optimization layers through differentiable optimization. In particular, by leveraging augmented Lagrangian techniques for solving QP layers that are potentially infeasible, we propose an extended conservative Jacobian formulation for differentiating convex QPs, covering both feasible and infeasible problems. For feasible problems, and when the solution is differentiable, this reduces to standard Jacobians. We further provide an open-source C++ framework, referred to as "QPLayer", which implements the approach. Through a classic learning example we have shown that differentiating over infeasible QP enables more structured learning with better predicting power. We have additionally proposed in the appendix more extensive benchmarks and experiments, to evaluate QPLayer speed and numerical robustness against other alternative state-of-the-art optimization layers. As for future plans, we will extend QPLayer to deal with a broader range of optimization layers that include second-order cones.

---

[3]Such behavior could be explained by the fact that the problem is harder to solve, and OptNet fails outputting accurate solutions. Furthermore, as it learns a strictly convex QP instead of a LP, the approximation of the model learned is less accurate for solving Sudoku, which are fundamentally formulated as MILP.

## ACKNOWLEDGMENTS

This work was supported in part by the French government under the management of Agence Nationale de la Recherche through the NIMBLE project (ANR-22-CE33-0008) and as part of the "Investissements d'avenir" program, reference ANR-19-P3IA-0001 (PRAIRIE 3IA Institute). This work was also supported by the European Union through the AGIMUS project (GA no.101070165), Louis Vuitton. This work has also been supported by the Paris Île-de-France Région in the framework of DIM AI4IDF. Views and opinions expressed are those of the author(s) only and do not necessarily reflect those of the European Union or the European Commission. Neither the European Union nor the European Commission can be held responsible for them.

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

ORGANIZATION OF THE APPENDIX

| Section | Content |
|---|---|
| Appendix A | Lemma 1: solutions to QP-H($\theta$). |
| Appendix B ECJs and automatic differentiation. | Lemma 2: path differentiability of $G$ (Appendix B.1). |
| | Forward AD (general case, Appendix B.2.1.1). |
| | Backward AD (general case, Appendix B.2.1.2). |
| | Forward AD (feasible QPs, Appendix B.2.2.1). |
| | Backward AD (structurally feasible QPs, Appendix B.2.2.2). |
| | Lemma 3: when do ECJs reduce to Jacobians? (Appendix B.3). |
| Appendix C.1 Pedagogical examples. | Strictly convex QP (parameterized constraints) (Appendix C.1.1). |
| | Strictly convex QP (parameterized objective) (Appendix C.1.2). |
| | Parameterized linear program (Appendix C.1.3). |
| Appendix C.2 | Additional benchmarks (timing). |
| Appendix C.3 | Training with large learning rates experiments. |
| Appendix D Experimental setups. | Layer architecture for the Sudoku problem (Appendix D.1). |
| | Description of the cart-pole problem (Appendix D.2). |

Table 1: Organization of the appendix. QP stands for "quadratic programming", AD stands for "automatic differentiation", and (E)CJ stands for (extended) conservative Jacobian.

## A  PROOF OF LEMMA 1

For proving Lemma 1, we first show that solutions to QP-H($\theta$) are zeros of the map $\mathcal{G}$:

$$\mathcal{G}(x, z; \theta) := \begin{bmatrix} H(\theta)x + g(\theta) + C(\theta)^\top z \\ [[C(\theta)x - u(\theta)]_- + z]_+ - z \\ C(\theta)^\top [C(\theta)x - u(\theta)]_+ \end{bmatrix}. \tag{12}$$

Then, a suitable change of variable shows that finding a zero of $\mathcal{G}$ is equivalent to finding a zero of map $G$.

**Lemma 1.** *Let $H(\theta) \in \mathcal{S}_+^n(\mathbb{R})$, $g(\theta) \in \mathbb{R}^n$, $C(\theta) \in \mathbb{R}^{n_i \times n}$ and $u(\theta) \in \mathbb{R}^{n_i}$ be satisfying Assumption 1. It holds that $(x^\star, z^\star, s^\star)$ solves QP-H($\theta$) iff there exists $t^\star \in \mathbb{R}^{n_i}$ s.t. $G(x^\star, z^\star, t^\star; \theta) = 0$ and $s^\star = [t^\star]_+$.*

*Proof.* We first show that $(x^\star, z^\star, [Cx^\star - u]_+)$ solves equation QP-H($\theta$) if and only if $\mathcal{G}(x^\star, z^\star; \theta) = 0$.

The optimal shift $s^\star$ (that corresponds to the closest feasible QP) is equal to $[Cx^\star - u]_+$ and is characterized by the $\ell_2$ optimality condition (Chiche & Gilbert, 2016, Lemma 2.13):

$$C^\top [Cx^\star - u]_+ = 0.$$

Furthermore, for a feasible problem, the KKT conditions using nonlinear complementarity formulation (Sun & Qi, 1999) reads (De Marchi, 2022, Section 2.1):

$$Hx^\star + g + C^\top z^\star = 0, \\ [Cx^\star - u + z^\star]_+ - z^\star = 0. \tag{13}$$

For showing equivalence, it is thereby sufficient to show that the second line of equation 13 corresponds to:

$$[[Cx^\star - u]_- + z^\star]_+ - z^\star = 0. \tag{14}$$

This equivalence is straightforward when equation QP($\theta$) is feasible, it therefore follows that we only need to handle the infeasible case. When equation QP($\theta$) is primal infeasible, then $t^\star = Cx^\star - u$ has a set of components $I \sqsubset [1, n_i]$ strictly positive, hence $s_I^\star = [t_I^\star]_+ = t_I^\star > 0$. For these components, a solution $x^\star$ of the closest feasible QP lies on the border $C_I x^\star = u_I + t_I^\star$. The complementarity condition (De Marchi, 2022, Section 2.1) for these components reads:

$$\underbrace{[Cx^\star - u_I - t_I^\star}_{=0} + z_I^\star]_+ - z_I^\star = 0,$$

and we thus have $[z_I^\star]_+ = z_I^\star$. For the other set of components, which we denote by $I^c$, we have that $s_{I^c}^\star = 0$, and hence $x^\star$ follows the complementary conditions as in the feasible case

$$[Cx^\star - u_I^c + z_{I^c}^\star]_+ - z_{I^c}^\star = 0.$$

Therefore, it follows that equation 14 captures the two cases (that is, both feasible and infeasible QPs), which concludes the first part of the proof.

Finally, introducing the slack variable $t^\star = C(\theta)x^\star - u(\theta)$, we have

$$\mathcal{G}(x^\star, z^\star;\, \theta) = 0 \tag{15}$$

$$\Leftrightarrow \begin{bmatrix} H(\theta)x^\star + g(\theta) + C(\theta)^\top z^\star \\ [[C(\theta)x^\star - u(\theta)]_- + z^\star]_+ - z^\star \\ C(\theta)^\top [C(\theta)x^\star - u(\theta)]_+ \end{bmatrix} = 0 \tag{16}$$

$$\Leftrightarrow \begin{bmatrix} H(\theta)x^\star + g(\theta) + C(\theta)^\top z^\star \\ C(\theta)x^\star - u(\theta) - t^\star \\ [[t^\star]_- + z^\star]_+ - z^\star \\ C(\theta)^\top [t^\star]_+ \end{bmatrix} = 0 \tag{17}$$

$$G(x^\star, z^\star, t^\star;\, \theta) = 0. \tag{18}$$

Hence $G(x^\star, z^\star, t^\star) = 0$ iff $(x^\star, z^\star, [t^\star]_+)$ solves QP-H($\theta$), which concludes.

$\square$

## B  ECJs AND AUTOMATIC DIFFERENTIATION

This section provides the proofs of the different results used in Section 3. In particular, we define ECJs and provide algorithms for computing them (both in forward and backward AD modes).

### B.1  PROOF OF LEMMA 2

**Lemma 2.** *G is path differentiable w.r.t. $x^\star$, $z^\star$ and $t^\star$. Furthermore, if $H(\theta)$, $g(\theta)$, $C(\theta)$ and $u(\theta)$ are differentiable w.r.t. $\theta$, then $G$ is path differentiable w.r.t. $\theta$.*

*Proof.* We start with the first claim of the lemma. The non-negative projector $[.]_+$ is (component-wise) convex, and hence path differentiable (Bolte & Pauwels, 2020, Proposition 2(i)). Thus, it remains to show that the third component of $G$ is path differentiable for reaching the desired conclusion.

To do so, we show that the third component is Lipschitz continuous and real semialgebraic (Bolte & Pauwels, 2020, Proposition 2(iv)). Without loss of generality, we restrict ourselves to the case with 2 components (one for the dual variables, and one for the slack variables) using the following function $h : \mathbb{R}^2 \to \mathbb{R}$, s.t. $h(z, s) := [[s]_- + z]_+ - z$. Then, the following Lipschitzness argument applies component-wise.

Let $(s_1, z_1) \in \mathbb{R}^2$ and $(s_2, z_2) \in \mathbb{R}^2$:

$$
\begin{aligned}
|h(z_1, s_1) - h(z_2, s_2)| &\leqslant |[[s_1]_- + z_1]_+ - [[s_2]_- + z_2]_+| + |z_1 - z_2| \\
&\leqslant |[s_1]_- + z_1 - [s_2]_- - z_2| + |z_1 - z_2| \text{ by monotonicity of } [.]_+ \\
&\leqslant |[s_1]_- - [s_2]_-| + 2|z_1 - z_2| \\
&= |s_1 - [s_1]_+ - (s_2 - [s_2]_+)| + 2|z_1 - z_2| \\
&\leqslant |s_1 - s_2| + |[s_1]_+ - [s_2]_+| + 2|z_1 - z_2| \\
&\leqslant 2|s_1 - s_2| + 2|z_1 - z_2| \text{ by monotonicity of } [.]_+ \\
&\leqslant 2\sqrt{2} \| \begin{bmatrix} s_1 \\ z_1 \end{bmatrix} - \begin{bmatrix} s_2 \\ z_2 \end{bmatrix} \|_2.
\end{aligned}
$$

Therefore, $h$ is Lipschitz continuous. For showing that the third component $G$ describes a semi-algebraic set, we explicitly formulate the graph of $h$ as a finite union of base semi-algebraic sets, as follows:

$$
\begin{aligned}
\text{gph}(h) = &\{(z, s, y) \in \mathbb{R}^3 | y = [s]_- \text{ and } [s]_- + z > 0\} \\
&\cup \{(z, s, y) \in \mathbb{R}^3 | y = -z \text{ and } [s]_- + z = 0\} \\
&\cup \{(z, s, y) \in \mathbb{R}^3 | y = -z \text{ and } [s]_- + z < 0\} \\
= &\{(z, s, y) \in \mathbb{R}^3 | y = s \text{ and } s + z > 0 \text{ and } s < 0\} \\
&\cup \{(z, s, y) \in \mathbb{R}^3 | y = s \text{ and } s = 0\} \\
&\cup \{(z, s, y) \in \mathbb{R}^3 | y = 0 \text{ and } z > 0 \text{ and } s > 0\} \\
&\cup \{(z, s, y) \in \mathbb{R}^3 | y = -z \text{ and } s + z = 0 \text{ and } s < 0\} \\
&\cup \{(z, s, y) \in \mathbb{R}^3 | y = -z \text{ and } z = 0 \text{ and } s = 0\} \\
&\cup \{(z, s, y) \in \mathbb{R}^3 | y = -z \text{ and } z = 0 \text{ and } s > 0\} \\
&\cup \{(z, s, y) \in \mathbb{R}^3 | y = -z \text{ and } s + z < 0 \text{ and } s < 0\} \\
&\cup \{(z, s, y) \in \mathbb{R}^3 | y = -z \text{ and } z < 0 \text{ and } s = 0\} \\
&\cup \{(z, s, y) \in \mathbb{R}^3 | y = -z \text{ and } z < 0 \text{ and } s > 0\}.
\end{aligned}
$$

Hence, $\text{gph}(h)$ is real and semi-algebraic as it is a finite union of sets defined by polynomial equalities and inequalities. Hence $h$ is Lipschitz continuous and real semi-algebraic, thereby reaching the target conclusion for the first part of Lemma 2.

As for the second part of Lemma 2. $G$ is linear, and hence differentiable, w.r.t. $H(\theta)$, $g(\theta)$, $C(\theta)$ and $u(\theta)$. Furthermore, by assumption $H(\theta)$, $g(\theta)$, $C(\theta)$ and $u(\theta)$ are differentiable w.r.t. $\theta$. As differentiability implies path-differentiability (Bolte & Pauwels, 2020, Remark 3b), we arrive at the desired claim by the conservativity of the chain rule for path-differentiable functions (Bolte & Pauwels, 2020, Proposition 2). □

## B.2 Forward and backward AD for computing ECJs

This section provides technical details for the computation of ECJs in forward and backward modes for both primal feasible and infeasible problems. We further include the proofs of **??** and Lemma 3.

### B.2.1 General case

**B.2.1.1 Forward pass.** Let $H(\theta)$, $g(\theta)$, $C(\theta)$ and $u(\theta)$ be differentiable w.r.t. $\theta$ and satisfying Assumption 1, and $x^\star, z^\star, t^\star$ s.t. $G(x^\star, z^\star, t^\star; \theta) = 0$.

As $G$ is path-differentiable w.r.t. $v^\star := (x^\star, z^\star, t^\star)$ (see Lemma 2), we have

$$
\begin{bmatrix} H & C^\top & 0 \\ C & 0 & -I \\ 0 & \Pi_1 - I & \Pi_1 \Pi_2 \\ 0 & 0 & C^\top \Pi_3 \end{bmatrix} \in \frac{\partial G(x^\star, z^\star, t^\star; \theta)}{\partial v^\star},
$$

for some $\Pi_1 \in \partial[.]_+([t^\star]_- + z^\star)$, $\Pi_2 \in \partial[.]_-(t^\star)$ and $\Pi_3 \in \partial[.]_+(t^\star)$.

Furthermore, as $G$ is linear w.r.t. $H(\theta)$, $g(\theta)$, $C(\theta)$ and $u(\theta)$ and $H(\theta)$, $g(\theta)$, $C(\theta)$ and $u(\theta)$ are differentiable w.r.t. $\theta$, the usual chain rule dictates that

$$\frac{\partial G(x^\star, z^\star, t^\star;\, \theta)}{\partial \theta} = \begin{bmatrix} \frac{\partial H}{\partial \theta} x^\star + \frac{\partial g}{\partial \theta} + \frac{\partial C}{\partial \theta}^\top z^\star \\ \frac{\partial C}{\partial \theta} x^\star - \frac{\partial u}{\partial \theta} \\ 0 \\ \frac{\partial C}{\partial \theta}^\top [t^\star]_+ \end{bmatrix}.$$

Finally, as $\partial [.]_+(0) = \partial [.]_-(0) = [0, 1]$, we make the following arbitrary choices in zeros

$$\begin{aligned} \Pi_1 &= I \text{ when } [t^\star]_- + z^\star = 0, \\ \Pi_2 &= I \text{ when } t^\star = 0, \\ \Pi_3 &= 0 \text{ when } t^\star = 0, \end{aligned} \tag{19}$$

so that $\Pi_3 = I - \Pi_2$. ECJs are thus retrieved as solutions to:

$$\frac{\partial x^\star}{\partial \theta}, \frac{\partial z^\star}{\partial \theta}, \frac{\partial t^\star}{\partial \theta} \in \underset{\frac{\partial x}{\partial \theta}, \frac{\partial z}{\partial \theta}, \frac{\partial t}{\partial \theta}}{\arg\min} \left\| \begin{bmatrix} H & C^\top & 0 \\ C & 0 & -I \\ 0 & \Pi_1 - I & \Pi_1 \Pi_2 \\ 0 & 0 & C^\top(I - \Pi_2) \end{bmatrix} \begin{bmatrix} \frac{\partial x}{\partial \theta} \\ \frac{\partial z}{\partial \theta} \\ \frac{\partial t}{\partial \theta} \end{bmatrix} + \begin{bmatrix} \frac{\partial H}{\partial \theta} x^\star + \frac{\partial g}{\partial \theta} + \frac{\partial C}{\partial \theta}^\top z^\star \\ \frac{\partial C}{\partial \theta} x^\star - \frac{\partial u}{\partial \theta} \\ 0 \\ \frac{\partial C}{\partial \theta}^\top [t^\star]_+ \end{bmatrix} \right\|_2^2. \tag{20}$$

In practice, solving this problem can be done via an augmented Lagrangian-based solver for the problem:

$$\frac{\partial x^\star}{\partial \theta}, \frac{\partial z^\star}{\partial \theta}, \frac{\partial t^\star}{\partial \theta} \in \underset{\frac{\partial x}{\partial \theta}, \frac{\partial z}{\partial \theta}, \frac{\partial t}{\partial \theta}}{\arg\min}\, 0$$
$$\text{s.t.} \begin{bmatrix} H & C^\top & 0 \\ C & 0 & -I \\ 0 & \Pi_1 - I & \Pi_1 \Pi_2 \\ 0 & 0 & C^\top(I - \Pi_2) \end{bmatrix} \begin{bmatrix} \frac{\partial x}{\partial \theta} \\ \frac{\partial z}{\partial \theta} \\ \frac{\partial t}{\partial \theta} \end{bmatrix} = - \begin{bmatrix} \frac{\partial H}{\partial \theta} x^\star + \frac{\partial g}{\partial \theta} + \frac{\partial C}{\partial \theta}^\top z^\star \\ \frac{\partial C}{\partial \theta} x^\star - \frac{\partial u}{\partial \theta} \\ 0 \\ \frac{\partial C}{\partial \theta}^\top [t^\star]_+ \end{bmatrix} \tag{21}$$

when this problem is feasible. When it is not feasible, the augmented Lagrangian naturally converges to the solution of the more general equation 20, see (Chiche & Gilbert, 2016, Proposition 4.2). In comparison to equation 20, a notable advantage of the formulation equation 21 is that it is naturally numerically more stable and sparse–by avoiding square matrix products from the objective.

#### B.2.1.2 Backward pass.
The following lemma formally details the results from Section 3.4.1.

**Lemma 4.** *Let $h : \mathbb{R}^n \times (\mathbb{R}^{n_i})^2 \to \mathbb{R}$ be a differentiable function, and let $H(\theta)$, $g(\theta)$, $C(\theta)$ and $u(\theta)$ be differentiable w.r.t. $\theta$ and satisfying Assumption 1. Then, denoting $\mathcal{L}(\theta) := h(x^\star(\theta), z^\star(\theta), s^\star(\theta))$ and under assumptions of Assumption 2 and Assumption 3, we have that $\frac{\partial \mathcal{L}}{\partial \theta}$ can be derived as follows*

$$\frac{\partial \mathcal{L}}{\partial \theta} = (b_1^\star)^\top \frac{\partial H}{\partial \theta} x^\star + (b_1^\star)^\top \frac{\partial g}{\partial \theta} + (b_2^\star)^\top \frac{\partial C}{\partial \theta} x^\star + (z^\star)^\top \frac{\partial C}{\partial \theta} b_1^\star + (s^\star)^\top \frac{\partial C}{\partial \theta} b_4^\star - (b_2^\star)^\top \frac{\partial u}{\partial \theta},$$

*where $b_1^\star$, $b_2^\star$, $b_3^\star$ and $b_4^\star$ are the solutions of the linear system*

$$\begin{bmatrix} H & C^\top & 0 & 0 \\ C & 0 & (I - \Pi_1) & 0 \\ 0 & -I & -\Pi_1 \Pi_2 & (1 - \Pi_2) C \end{bmatrix} \begin{bmatrix} b_1^\star \\ b_2^\star \\ b_3^\star \\ b_4^\star \end{bmatrix} = - \begin{bmatrix} \frac{\delta \mathcal{L}}{\delta x^\star} \\ \frac{\delta \mathcal{L}}{\delta z^\star} \\ \frac{\delta \mathcal{L}}{\delta s^\star} \end{bmatrix}.$$

*Proof.* Under the assumption of Assumption 2, it holds that $\begin{bmatrix} \frac{\partial x^\star}{\partial \theta} \\ \frac{\partial z^\star}{\partial \theta} \\ \frac{\partial s^\star}{\partial \theta} \end{bmatrix}$ is a CJ as $\begin{bmatrix} \frac{\partial x^\star}{\partial \theta} \\ \frac{\partial z^\star}{\partial \theta} \\ \frac{\partial t^\star}{\partial \theta} \end{bmatrix}$ and $(1 - \Pi_2) \frac{\partial t^\star}{\partial \theta} \in \frac{\partial s^\star}{\partial \theta}$ are CJs ((Bolte & Pauwels, 2020, Proposition 2)). Furthermore, as $\mathcal{L}$ is differentiable w.r.t. $x^\star$, $z^\star$ and $s^\star$, it is path-differentiable (Bolte & Pauwels, 2020, Remark 3b) w.r.t. $x^\star, z^\star, s^\star$, so

we can apply chain rule ((Bolte & Pauwels, 2020, Proposition 2)):

$$
\begin{aligned}
\frac{\partial \mathcal{L}}{\partial \theta} &= \frac{\partial \mathcal{L}}{\partial x^\star}^\top \frac{\partial x^\star}{\partial \theta} + \frac{\partial \mathcal{L}}{\partial z^\star}^\top \frac{\partial z^\star}{\partial \theta} + \frac{\partial \mathcal{L}}{\partial s^\star}^\top \frac{\partial s^\star}{\partial \theta} \\
&= -(-\begin{bmatrix} \frac{\delta \mathcal{L}}{\delta x^\star} \\ \frac{\delta \mathcal{L}}{\delta z^\star} \\ \frac{\delta \mathcal{L}}{\delta s^\star} \end{bmatrix})^\top \begin{bmatrix} \frac{\partial x^\star}{\partial \theta} \\ \frac{\partial z^\star}{\partial \theta} \\ \frac{\partial s^\star}{\partial \theta} \end{bmatrix} \\
&= -(\begin{bmatrix} H & C^\top & 0 & 0 \\ C & 0 & (I - \Pi_1) & 0 \\ 0 & -I & -\Pi_1 \Pi_2 & (1 - \Pi_2)C \end{bmatrix} \begin{bmatrix} b_1^\star \\ b_2^\star \\ b_3^\star \\ b_4^\star \end{bmatrix})^\top \begin{bmatrix} \frac{\partial x^\star}{\partial \theta} \\ \frac{\partial z^\star}{\partial \theta} \\ \frac{\partial s^\star}{\partial \theta} \end{bmatrix} \\
&= -\begin{bmatrix} b_1^\star \\ b_2^\star \\ b_3^\star \\ b_4^\star \end{bmatrix}^\top (-\begin{bmatrix} \frac{\partial H}{\partial \theta} x^\star + \frac{\partial g}{\partial \theta} + \frac{\partial C}{\partial \theta}^\top z^\star \\ \frac{\partial C}{\partial \theta} x^\star - \frac{\partial u}{\partial \theta} \\ 0 \\ \frac{\partial C}{\partial \theta}^\top [t^\star]_+ \end{bmatrix}) \\
&= (b_1^\star)^\top (\frac{\partial H}{\partial \theta} x^\star + \frac{\partial g}{\partial \theta} + \frac{\partial C}{\partial \theta}^\top z^\star) \\
&\quad + (b_2^\star)^\top (\frac{\partial C}{\partial \theta} x^\star - \frac{\partial u}{\partial \theta}) \\
&\quad + (b_4^\star)^\top (\frac{\partial C}{\partial \theta}^\top [t^\star]_+) \\
&= (b_1^\star)^\top \frac{\partial H}{\partial \theta} x^\star + (\frac{\partial C}{\partial \theta} b_1^\star)^\top z^\star + (b_1)^\top \frac{\partial g}{\partial \theta} \\
&\quad + (b_2^\star)^\top (\frac{\partial C}{\partial \theta} x^\star - \frac{\partial u}{\partial \theta}) \\
&\quad + (\frac{\partial C}{\partial \theta} b_4^\star)^\top [t^\star]_+).
\end{aligned}
$$

To conclude, we show there always exist solution $b_1^\star, b_2^\star, b_4^\star$ of the system:

$$
\begin{bmatrix} H & C^\top & 0 & 0 \\ C & 0 & (I - \Pi_1) & 0 \\ 0 & -I & -\Pi_1 \Pi_2 & (1 - \Pi_2)C \end{bmatrix} \begin{bmatrix} b_1^\star \\ b_2^\star \\ b_3^\star \\ b_4^\star \end{bmatrix} = -\begin{bmatrix} \frac{\delta \mathcal{L}}{\delta x^\star} \\ \frac{\delta \mathcal{L}}{\delta z^\star} \\ \frac{\delta \mathcal{L}}{\delta s^\star} \end{bmatrix}, \tag{22}
$$

Under Assumption 3, $\Pi_1 = I$ and $\Pi_2 = 0$ (since $s^\star = [t^\star]_+ > 0$). Furthermore, since $\frac{\delta \mathcal{L}}{\delta z^\star} = 0$, equation 22 is thus equivalent to solving

$$
\begin{bmatrix} H & C^\top & 0 & 0 \\ C & 0 & 0 & 0 \\ 0 & -I & 0 & C \end{bmatrix} \begin{bmatrix} b_1^\star \\ b_2^\star \\ b_3^\star \\ b_4^\star \end{bmatrix} = -\begin{bmatrix} \frac{\delta \mathcal{L}}{\delta x^\star} \\ 0 \\ \frac{\delta \mathcal{L}}{\delta s^\star} \end{bmatrix}.
$$

The latter system can be reformulated as

$$
C^\top C b_4^\star = \frac{\delta \mathcal{L}}{\delta x^\star} - C^\top \frac{\delta \mathcal{L}}{\delta s^\star}, \tag{23}
$$

$$
b_2^\star = \frac{\delta \mathcal{L}}{\delta s^\star} + C b_4^\star, \tag{24}
$$

$$
C b_1^\star = 0, \tag{25}
$$

which has always solutions since $C$ is supposed to be full row rank. $\square$

### B.2.2 SIMPLIFICATION OF WHEN QP IS FEASIBLE

**B.2.2.1 Simplification of the forward pass** When QP$(\theta)$ is feasible and $H(\theta)$, $g(\theta)$, $C(\theta)$ and $u(\theta)$ are differentiable w.r.t. $\theta$ and satisfy Assumption 1, then $[t^\star]_+ = 0$. Further, our choices of

subgradients at zero (see equation 19) imply that $\Pi_2 = I$ and hence the following simplifications

$$\begin{bmatrix} H & C^\top & 0 \\ C & 0 & -I \\ 0 & \Pi_1 - I & \Pi_1 \\ 0 & 0 & 0 \end{bmatrix} \in \frac{\partial G(x^\star, z^\star, t^\star;\ \theta)}{\partial v^\star},$$

$$\begin{bmatrix} \frac{\partial H}{\partial \theta} x^\star + \frac{\partial g}{\partial \theta} + \frac{\partial C}{\partial \theta}^\top z^\star \\ \frac{\partial C}{\partial \theta} x^\star - \frac{\partial u}{\partial \theta} \\ 0 \\ 0 \end{bmatrix} = \frac{\partial G(x^\star, z^\star, t^\star;\ \theta)}{\partial \theta}.$$

Moreover, the optimality conditions of

$$\frac{\partial x^\star}{\partial \theta}, \frac{\partial z^\star}{\partial \theta}, \frac{\partial t^\star}{\partial \theta} \in \operatorname*{arg\,min}_{\frac{\partial x}{\partial \theta}, \frac{\partial z}{\partial \theta}, \frac{\partial t}{\partial \theta}} \left\| \begin{bmatrix} H & C^\top & 0 \\ C & 0 & -I \\ 0 & \Pi_1 - I & \Pi_1 \\ 0 & 0 & 0 \end{bmatrix} \begin{bmatrix} \frac{\partial x}{\partial \theta} \\ \frac{\partial z}{\partial \theta} \\ \frac{\partial t}{\partial \theta} \end{bmatrix} + \begin{bmatrix} \frac{\partial H}{\partial \theta} x^\star + \frac{\partial g}{\partial \theta} + \frac{\partial C}{\partial \theta}^\top z^\star \\ \frac{\partial C}{\partial \theta} x^\star - \frac{\partial u}{\partial \theta} \\ 0 \\ 0 \end{bmatrix} \right\|^2_2,$$

write down

$$(H^2 + C^\top C)\frac{\partial x^\star}{\partial \theta} + HC^\top \frac{\partial z^\star}{\partial \theta} - C^\top \frac{\partial t^\star}{\partial \theta} + [H(\frac{\partial H}{\partial \theta} x^\star + \frac{\partial g}{\partial \theta} + \frac{\partial C}{\partial \theta}^\top z^\star) + C^\top(\frac{\partial C}{\partial \theta} x^\star - \frac{\partial u}{\partial \theta})] = 0,$$

$$CH\frac{\partial x^\star}{\partial \theta} + (C^\top C + I - \Pi_1)\frac{\partial z^\star}{\partial \theta} + C(\frac{\partial H}{\partial \theta} x^\star + \frac{\partial g}{\partial \theta} + \frac{\partial C}{\partial \theta}^\top z^\star) = 0, \quad (26)$$

$$-C\frac{\partial x^\star}{\partial \theta} + (I + \Pi_1)\frac{\partial t^\star}{\partial \theta} - (\frac{\partial C}{\partial \theta} x^\star - \frac{\partial u}{\partial \theta}) = 0.$$

Third equation of equation 26 leads to

$$\frac{\partial t^\star}{\partial \theta} = \frac{1}{1 + \Pi_1}(C\frac{\partial x^\star}{\partial \theta} + (\frac{\partial C}{\partial \theta} x^\star - \frac{\partial u}{\partial \theta})).$$

Hence, optimality conditions without variable $\frac{\partial t^\star}{\partial \theta}$ reduce to

$$(H^2 + C^\top \frac{\Pi_1}{1 + \Pi_1} C)\frac{\partial x^\star}{\partial \theta} + HC^\top \frac{\partial z^\star}{\partial \theta} + C^\top \frac{\Pi_1}{1 + \Pi_1}(\frac{\partial C}{\partial \theta} x^\star - \frac{\partial u}{\partial \theta}) + H(\frac{\partial H}{\partial \theta} x^\star + \frac{\partial g}{\partial \theta} + \frac{\partial C}{\partial \theta}^\top z^\star) = 0,$$

$$(27)$$

$$CH\frac{\partial x^\star}{\partial \theta} + (C^\top C + I - \Pi_1)\frac{\partial z^\star}{\partial \theta} + C(\frac{\partial H}{\partial \theta} x^\star + \frac{\partial g}{\partial \theta} + \frac{\partial C}{\partial \theta}^\top z^\star) = 0.$$

Furthermore, the following problem

$$\min_{\frac{\partial x}{\partial \theta}, \frac{\partial z}{\partial \theta}} \left\| \begin{bmatrix} H & C^\top \\ \frac{\Pi_1}{\sqrt{1+\Pi_1}} C & \Pi_1 - I \end{bmatrix} \begin{bmatrix} \frac{\partial x}{\partial \theta} \\ \frac{\partial z}{\partial \theta} \end{bmatrix} + \begin{bmatrix} \frac{\partial H}{\partial \theta} x^\star + \frac{\partial g}{\partial \theta} + \frac{\partial C}{\partial \theta}^\top z^\star \\ \frac{\Pi_1}{\sqrt{1+\Pi_1}}(\frac{\partial C}{\partial \theta} x^\star - \frac{\partial u}{\partial \theta}) \end{bmatrix} \right\|^2_2,$$

have the same KKT conditions as equation 27, thereby allowing to simplify the problem as follows:

$$\min_{\frac{\partial x}{\partial \theta}, \frac{\partial z}{\partial \theta}, \frac{\partial t}{\partial \theta}} \left\| \frac{\partial G(x^\star, z^\star, s^\star;\ \theta)}{\partial \hat{v}^\star} \begin{bmatrix} \frac{\partial x}{\partial \theta} \\ \frac{\partial z}{\partial \theta} \\ \frac{\partial t}{\partial \theta} \end{bmatrix} + \frac{\partial G(x^\star, z^\star, s^\star;\ \theta)}{\partial \theta} \right\|^2_2$$

$$(28)$$

$$= \min_{\frac{\partial x}{\partial \theta}, \frac{\partial z}{\partial \theta}} \left\| \begin{bmatrix} H & C^\top \\ \frac{\Pi_1}{\sqrt{I+\Pi_1}} C & \Pi_1 - I \end{bmatrix} \begin{bmatrix} \frac{\partial x}{\partial \theta} \\ \frac{\partial z}{\partial \theta} \end{bmatrix} + \begin{bmatrix} \frac{\partial H}{\partial \theta} x^\star + \frac{\partial g}{\partial \theta} + \frac{\partial C}{\partial \theta}^\top z^\star \\ \frac{\Pi_1}{\sqrt{1+\Pi_1}}(\frac{\partial C}{\partial \theta} x^\star - \frac{\partial u}{\partial \theta}) \end{bmatrix} \right\|^2_2,$$

Hence

$$\frac{\partial x^\star}{\partial \theta}, \frac{\partial z^\star}{\partial \theta}, \frac{\partial t^\star}{\partial \theta} \in \operatorname*{arg\,min}_{\frac{\partial x}{\partial \theta}, \frac{\partial z}{\partial \theta}, \frac{\partial t}{\partial \theta}} \left\| \frac{\partial G(x^\star, z^\star, s^\star;\ \theta)}{\partial \hat{v}^\star} \begin{bmatrix} \frac{\partial x}{\partial \theta} \\ \frac{\partial z}{\partial \theta} \\ \frac{\partial t}{\partial \theta} \end{bmatrix} + \frac{\partial G(x^\star, z^\star, s^\star;\ \theta)}{\partial \theta} \right\|^2_2$$

is equivalent to

$$\frac{\partial x^\star}{\partial \theta}, \frac{\partial z^\star}{\partial \theta} \in \operatorname*{arg\,min}_{\frac{\partial x}{\partial \theta}, \frac{\partial z}{\partial \theta}} \left\| \begin{bmatrix} H & C^\top \\ \frac{\Pi_1}{\sqrt{1+\Pi_1}} C & \Pi_1 - I \end{bmatrix} \begin{bmatrix} \frac{\partial x}{\partial \theta} \\ \frac{\partial z}{\partial \theta} \end{bmatrix} + \begin{bmatrix} \frac{\partial H}{\partial \theta} x^\star + \frac{\partial g}{\partial \theta} + \frac{\partial C}{\partial \theta}^\top z^\star \\ \frac{\Pi_1}{\sqrt{1+\Pi_1}}(\frac{\partial C}{\partial \theta} x^\star - \frac{\partial u}{\partial \theta}) \end{bmatrix} \right\|^2_2,$$

$$\frac{\delta t^\star}{\delta \theta} = (I + \Pi_1)^{-1}(C\frac{\delta x^\star}{\delta \theta} + \frac{\partial C}{\partial \theta} x^\star - \frac{\partial u}{\partial \theta}).$$

**B.2.2.2 Simplification of the backward pass.** This sections details the results from Section 3.4.2.

**Lemma 5.** *Let $h : \mathbb{R}^n \times (\mathbb{R}^{n_i}) \to \mathbb{R}$ be a differentiable function, and let $H(\theta)$, $g(\theta)$, $C(\theta)$ and $u(\theta)$ be differentiable w.r.t. $\theta$ and satisfying Assumption 1. Then, denoting $\mathcal{L}(\theta) := h(x^\star(\theta), z^\star(\theta))$, we have under assumptions of Lemma 3 that $\frac{\partial \mathcal{L}}{\partial \theta}$ can be derived as follows*

$$\frac{\partial \mathcal{L}}{\partial \theta} = (b_x^\star)^\top \frac{\partial H}{\partial \theta} x^\star + (b_x^\star)^\top \frac{\partial g}{\partial \theta} + (\Pi_1 b_z^\star)^\top \frac{\partial C}{\partial \theta} x^\star + (z^\star)^\top \frac{\partial C}{\partial \theta} b_x^\star - (\Pi_1 b_z^\star)^\top \frac{\partial u}{\partial \theta},$$

*with $b_x^\star$, $b_z^\star$, the solution of the following linear system*

$$\begin{bmatrix} H & C^\top \Pi_1 \\ C & -(I - \Pi_1) \end{bmatrix} \begin{bmatrix} b_x \\ b_z \end{bmatrix} = - \begin{bmatrix} \frac{\delta \mathcal{L}}{\delta x^\star} \\ \frac{\delta \mathcal{L}}{\delta z^\star} \end{bmatrix},$$

*Furthermore, this latter linear system can be solved using iterative refinement.*

*Proof.* Under the assumptions of Lemma 3, it holds that $\begin{bmatrix} \frac{\partial x^\star}{\partial \theta} \\ \frac{\partial z^\star}{\partial \theta} \end{bmatrix}$ is a Jacobian. Furthermore, as $\mathcal{L}$ is differentiable, the chain rule implies that:

$$\begin{aligned}
\frac{\partial \mathcal{L}}{\partial \theta} &= \begin{bmatrix} \frac{\partial \mathcal{L}}{\partial x^\star} \\ \frac{\partial \mathcal{L}}{\partial z^\star} \end{bmatrix}^\top \begin{bmatrix} \frac{\partial x^\star}{\partial \theta} \\ \frac{\partial z^\star}{\partial \theta} \end{bmatrix} \\
&= -(-\begin{bmatrix} \frac{\partial \mathcal{L}}{\partial x^\star} \\ \frac{\partial \mathcal{L}}{\partial z^\star} \end{bmatrix})^\top \begin{bmatrix} \frac{\partial x^\star}{\partial \theta} \\ \frac{\partial z^\star}{\partial \theta} \end{bmatrix} \\
&= -(\begin{bmatrix} H & C^\top \\ \Pi_1 C & \Pi_1 - I \end{bmatrix}^\top \begin{bmatrix} b_x \\ b_z \end{bmatrix})^\top \begin{bmatrix} \frac{\partial x^\star}{\partial \theta} \\ \frac{\partial z^\star}{\partial \theta} \end{bmatrix} \text{ as the matrix is nonsingular (see Lemma 3)} \\
&= -\begin{bmatrix} b_x \\ b_z \end{bmatrix}^\top (-\begin{bmatrix} \frac{\partial H}{\partial \theta} x^\star + \frac{\partial g}{\partial \theta} + \frac{\partial C}{\partial \theta}^\top z^\star \\ \Pi_1 (\frac{\partial C}{\partial \theta} x^\star - \frac{\partial u}{\partial \theta}) \end{bmatrix}) \\
&= (b_x)^\top (\frac{\partial H}{\partial \theta} x^\star + \frac{\partial g}{\partial \theta} + \frac{\partial C}{\partial \theta}^\top z^\star) \\
&\quad + (b_z)^\top (\Pi (\mathrm{d}C x^\star - \mathrm{d}u)) \\
&= [(b_x)^\top \frac{\partial H}{\partial \theta} x^\star + (\frac{\partial C}{\partial \theta} b_x)^\top z^\star + (b_x)^\top \frac{\partial g}{\partial \theta}] \\
&\quad + (b_z)^\top (\Pi (\frac{\partial C}{\partial \theta} x^\star - \frac{\partial u}{\partial \theta})),
\end{aligned}$$

where $(b_x, b_z)$ is a solution to

$$\begin{bmatrix} H & C^\top \Pi_1 \\ C & -(I - \Pi_1) \end{bmatrix} \begin{bmatrix} b_x \\ b_z \end{bmatrix} = - \begin{bmatrix} \frac{\delta \mathcal{L}}{\delta x^\star} \\ \frac{\delta \mathcal{L}}{\delta z^\star} \end{bmatrix}.$$

As detailed in the proof of Lemma 3 (see details in Appendix B.3), one can equivalently solve

$$\begin{bmatrix} H & C_J^\top \\ C_J & 0 \end{bmatrix} \begin{bmatrix} b_x \\ b_{z_J} \end{bmatrix} = - \begin{bmatrix} \frac{\delta \mathcal{L}}{\delta x^\star} \\ \frac{\delta \mathcal{L}}{\delta z_J^\star} \end{bmatrix},$$

$$b_{z_J^c} = \frac{\delta \mathcal{L}}{\delta z_{J^c}^\star},$$

with $J^c$ the index set for which the solution is strictly feasible (i.e., $i \in [1, n_i]$, $(\Pi_1)_i = 0$), and $J$ the set of active constraints (i.e., for which $(\Pi_1)_i = 1$). Such linear systems can be solved e.g., via iterative refinement (as the matrix involved is symmetric positive semi-definite (Parikh & Boyd, 2014, Section 4.1.2)). $\qquad\square$

## B.3 PROOF OF LEMMA 3

This section details the proof of Lemma 3, ensuring that, under some regularity assumptions, ECJs reduce to standard Jacobians.

**Lemma 3.** *If $QP(\theta)$ is feasible and $H(\theta)$, $g(\theta)$, $C(\theta)$ and $u(\theta)$ are differentiable w.r.t. $\theta$ and satisfy Assumption 1, and if the KKT matrix of active constraints is nonsingular and $x^\star$, $z^\star$ satisfy strict complementarity, then the ECJs matches the standard Jacobian, i.e., $\frac{\partial x^\star(\theta)}{\partial \theta} = \nabla x^\star(\theta)$ and $\frac{\partial z^\star(\theta)}{\partial \theta} = \nabla z^\star(\theta)$.*

*Proof.* If equation $QP(\theta)$ is feasible, then the ECJs of $x^\star$ and $z^\star$ w.r.t. $\theta$ are provided by

$$\frac{\partial x^\star}{\partial \theta}, \frac{\partial z^\star}{\partial \theta} \in \arg\min_{\frac{\partial x}{\partial \theta}, \frac{\partial z}{\partial \theta}} \left\| \underbrace{\begin{bmatrix} H & C^\top \\ \frac{\Pi_1}{\sqrt{1+\Pi_1}}C & \Pi_1 - I \end{bmatrix}}_{:=\Delta} \begin{bmatrix} \frac{\partial x}{\partial \theta} \\ \frac{\partial z}{\partial \theta} \end{bmatrix} + \begin{bmatrix} \frac{\partial H}{\partial \theta}x^\star + \frac{\partial g}{\partial \theta} + \frac{\partial C}{\partial \theta}^\top z^\star \\ \frac{\Pi_1}{\sqrt{1+\Pi_1}}(\frac{\partial C}{\partial \theta}x^\star - \frac{\partial u}{\partial \theta}) \end{bmatrix} \right\|_2^2, \quad (29)$$

where $\Pi_1$ corresponds to a binary diagonal matrix of the complementarity conditions $Cx^\star - u + z^\star \geqslant 0$. Denoting by $J^c$ the index set for which the solution is strictly feasible (i.e., $i \in [1, n_i]$, $(\Pi_1)_i = 0$), and by $J$ the index set of active constraints (i.e., for which $(\Pi_1)_i = 1$) then $\Delta$ can be reformulated as follows (by strict complementary)

$$\Delta = \begin{bmatrix} H & C_J^\top & C_{J^c}^\top \\ \frac{1}{\sqrt{2}}C_J & 0 & 0 \\ 0 & 0 & -I \end{bmatrix},$$

with $I$ being the identity matrix of appropriate dimension. Furthermore, the right-hand side of the linear system within the $\ell_2$ norm becomes

$$\begin{bmatrix} \frac{\partial H}{\partial \theta}x^\star + \frac{\partial g}{\partial \theta} + \frac{\partial C}{\partial \theta}^\top z^\star \\ \frac{1}{\sqrt{2}}(\frac{\partial C_J}{\partial \theta}x^\star - \frac{\partial u_J}{\partial \theta}) \\ 0 \end{bmatrix}.$$

$\begin{bmatrix} H & C_J^\top \\ C_J & 0 \end{bmatrix}$ corresponds to the KKT matrix of active constraints, and is nonsingular by assumption. As it implies nonsingularity of $\begin{bmatrix} H & C_J^\top \\ \frac{1}{\sqrt{2}}C_J & 0 \end{bmatrix}$, it follows that :

$$\frac{\partial z_{J^c}^\star}{\partial \theta} = 0,$$

and the solution to equation 29 is uniquely determined as the solution of the following linear system (as in (Amos & Kolter, 2017, Appendix A), after multiplying second row block by $\sqrt{2}$):

$$\begin{bmatrix} H & C_J^\top \\ C_J & 0 \end{bmatrix} \begin{bmatrix} \frac{\partial x^\star}{\partial \theta} \\ \frac{\partial z^\star}{\partial \theta} \end{bmatrix} = - \begin{bmatrix} \frac{\partial H}{\partial \theta}x^\star + \frac{\partial g}{\partial \theta} + \frac{\partial C}{\partial \theta}^\top z^\star \\ \frac{\partial C_J}{\partial \theta}x^\star - \frac{\partial u_J}{\partial \theta} \end{bmatrix},$$

Hence, we arrive at the desired conclusion that ECJ coincides with the usual Jacobian in this case.

$\square$

## C ADDITIONAL EXPERIMENTAL RESULTS

Appendix C.1 provides a few simple experiments with parametric QPs to illustrate the concept of ECJs. Appendix C.2 contains additional benchmarks. Appendix C.3 illustrates through several experiments that QPLayer can be trained with large learning rates.

### C.1 PEDAGOGICAL EXAMPLES OF PARAMETRIC QPS

A few numerical examples illustrate the concept of ECJ in different simple scenarios. The first example corresponds to a strictly convex parametric QP which can be either feasible or infeasible. In this example, a linear constraint depends on a parameter $\theta$. Depending on the value of this parameter, the QP can either be feasible or infeasible.

The second example is a strictly convex QP with a parameterized linear cost. This problem is always feasible.

The last example is a parametric LP with possibly multiple solutions. For appropriate values of the parameters, the LP is feasible but not differentiable.

#### C.1.1 STRICTLY CONVEX QP (PARAMETERIZED CONSTRAINTS)

Consider the following strictly convex QP parameterized by a scalar value $\theta$

$$
\begin{aligned}
x^\star(\theta) = \underset{x_1, x_2 \in \mathbb{R}^2}{\arg\min} & \frac{1}{2}(x_1^2 + x_2^2) \\
\text{s.t. } \theta \leqslant & x_1 + x_2 \leqslant 1.55, \\
1.5 \leqslant & 2x_1 + x_2 \leqslant 1.55
\end{aligned}
\tag{30}
$$

Notice that for $\theta > \theta_{\text{limit}} := 1.55$, the QP becomes primal infeasible. We use gradient descent to minimize two scalar losses $\mathcal{L}_1(\theta) = x_1^\star(\theta)$ and $\mathcal{L}_2(\theta) = x_2^\star(\theta)$, starting from a predefined value $\theta_0$. More precisely we have launched gradient descent for 40 steps with a learning rate $5 \times 10^{-4}$ starting from $\theta_0 = 1.54$. Figure 6a illustrates the results by showing the iterates of gradient descent for minimizing $x_1^\star(\theta)$ (as well as the search direction—minus the ECJs). By doing so $\theta$ increases and eventually becomes larger than $\theta_{\text{limit}}$.

Figure 6b reports a similar experiment for when minimizing $x_2^\star(\theta)$.

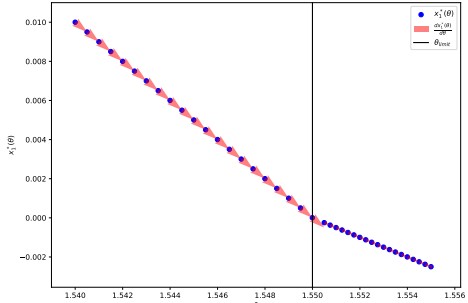

(a) 40 steps of gradient descent for minimizing $x_1^\star(\theta)$ starting from $\theta_0 = 1.54$. When $\theta > 1.55$, equation 30 is differentiated though infeasible.

(b) 40 steps of gradient descent for minimizing $x_2^\star(\theta)$ starting from $\theta_0 = 1.54$. The QPs remain feasible.

#### C.1.2 STRICTLY CONVEX QP (PARAMETERIZED OBJECTIVE)

Consider the following strictly convex QP parametrized by a scalar value $\theta$

$$
\begin{aligned}
x^\star(\theta) = \underset{x_1, x_2 \in \mathbb{R}^2}{\arg\min} & \frac{1}{2} \left\| \begin{bmatrix} x_1 \\ x_2 \end{bmatrix} + \begin{bmatrix} \theta \\ -2 \end{bmatrix} \right\|_2^2 \\
\text{s.t. } -300 \leqslant & x_1 + x_2 \leqslant 400, \\
-200 \leqslant & 2x_1 + x_2 \leqslant 500
\end{aligned}
\tag{31}
$$

We use gradient descent to minimize the loss $\mathcal{L}_1(\theta) = x_1^\star(\theta)$. More precisely, we run 40 iterations of gradient descent with learning rate $5 \times 10^{-4}$ starting from $\theta_0 = 1.54$, as reported by Figure 7. As expected, we see that $x_1^\star(\theta) = -\theta$, hence increasing $\theta$ decreases $x_1^\star(\theta)$.

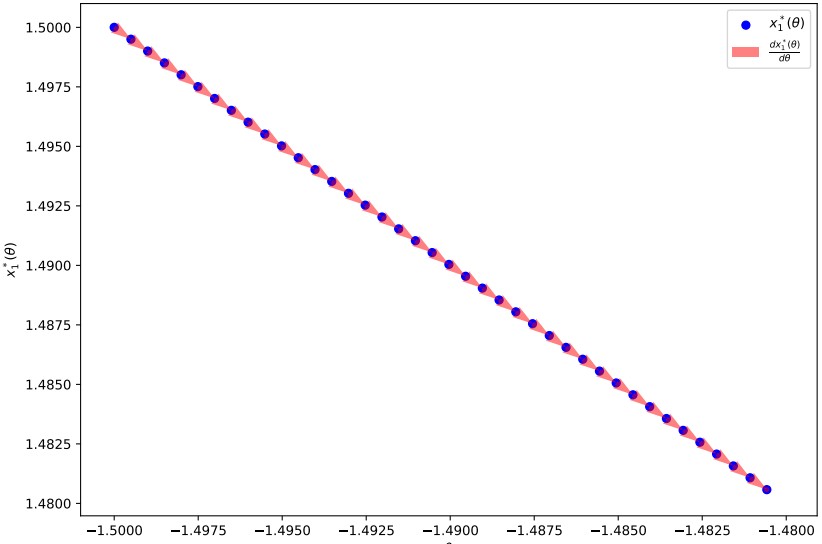

Figure 7: Gradient descent for minimizing $x_1^\star(\theta)$ (solution to equation 31) starting from $\theta_0 = -1.5$.

### C.1.3 PARAMETERIZED LINEAR PROGRAM

Consider the following LP parameterized by a scalar parameter $\theta > 0$

$$
\begin{aligned}
x^\star(\theta) \in \underset{x_1, x_2 \in \mathbb{R}^2}{\arg\min} \; & x_1 + x_2 \\
\text{s.t. } & \theta \leqslant x_1 + x_2, \\
& 0 \leqslant x_1 \leqslant 1, \\
& 0 \leqslant x_2 \leqslant 1.
\end{aligned}
\tag{32}
$$

Note that this LP is always well-defined for any $\theta$ since the linear cost is orthogonal to the recession cone (which is empty), thereby satisfying the technical requirements from Assumption 1. We use gradient descent for minimizing two scalar losses $\mathcal{L}_1(\theta) = x_1^\star(\theta)$ and $\mathcal{L}_2(\theta) = x_2^\star(\theta)$.

**C.1.3.1 Feasible case ($\theta \leqslant 2$).** For any $\theta \in ]0, 2]$, there are infinitely many solutions to equation 32 which are defined by the segment equation

$$
\begin{aligned}
x_1^\star + x_2^\star &= \theta, \\
0 \leqslant x_1^\star &\leqslant 1, \\
0 \leqslant x_2^\star &\leqslant 1.
\end{aligned}
$$

We can see in Figure 8a and Figure 8b that the forward pass chooses as solution $x_1^\star = x_2^\star = \frac{\theta}{2}$. Hence, only the constraint $\theta \leqslant x_1 + x_2$ is active. Following the formalism from Section 3.1 we have

$$
C = \begin{bmatrix} -1 & -1 \\ 1 & 0 \\ -1 & 0 \\ 0 & 1 \\ 0 & -1 \end{bmatrix}, \qquad\qquad u = \begin{bmatrix} -\theta \\ 1 \\ 0 \\ 1 \\ 0 \end{bmatrix}.
$$

The ECJs of $\mathcal{L}_1$ and $\mathcal{L}_2$ w.r.t $\theta$ are the solutions to

$$
\begin{bmatrix} (b_x^\star)_1 \\ (b_x^\star)_2 \\ b_z^\star \end{bmatrix} \in \arg\min_{b_x, b_z} \left\| \begin{bmatrix} 0 & 0 & -1 \\ 0 & 0 & -1 \\ -1 & -1 & 0 \end{bmatrix} \begin{bmatrix} (b_x)_1 \\ (b_x)_2 \\ b_z \end{bmatrix} + \begin{bmatrix} 1 \\ 0 \\ 0 \end{bmatrix} \right\|_2^2,
$$

$$
\begin{bmatrix} (d_x^\star)_1 \\ (d_x^\star)_2 \\ d_z^\star \end{bmatrix} \in \arg\min_{d_x, d_z} \left\| \begin{bmatrix} 0 & 0 & -1 \\ 0 & 0 & -1 \\ -1 & -1 & 0 \end{bmatrix} \begin{bmatrix} (d_x)_1 \\ (d_x)_2 \\ d_z \end{bmatrix} + \begin{bmatrix} 0 \\ 1 \\ 0 \end{bmatrix} \right\|_2^2.
$$

As the corresponding linear systems involved within the $\ell_2$ norm are infeasible, the least square estimates do not correspond to solutions to the linear system. That is, the corresponding least-square solutions are respectively the solutions of the following projected linear systems:

$$
\begin{bmatrix} 1 & 1 & 0 \\ 1 & 1 & 0 \\ 0 & 0 & 2 \end{bmatrix} \begin{bmatrix} (b_x^\star)_1 \\ (b_x^\star)_2 \\ b_z^\star \end{bmatrix} = \begin{bmatrix} 0 \\ 0 \\ 1 \end{bmatrix},
$$

$$
\begin{bmatrix} 1 & 1 & 0 \\ 1 & 1 & 0 \\ 0 & 0 & 2 \end{bmatrix} \begin{bmatrix} (d_x^\star)_1 \\ (d_x^\star)_2 \\ d_z^\star \end{bmatrix} = \begin{bmatrix} 0 \\ 0 \\ 1 \end{bmatrix},
$$

which leads to $\frac{\partial \mathcal{L}_1}{\partial \theta} = \frac{\partial \mathcal{L}_2}{\partial \theta} = b_z^\star = d_z^\star = \frac{1}{2}$. Figure 8a and Figure 8b show that those directions allow minimizing $\mathcal{L}_1$ and $\mathcal{L}_2$ through gradient descent.

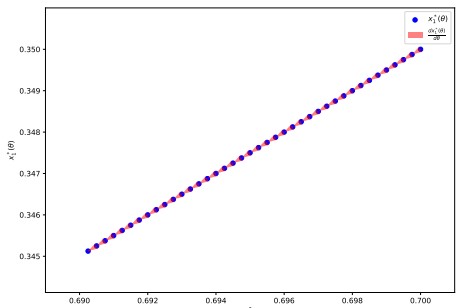

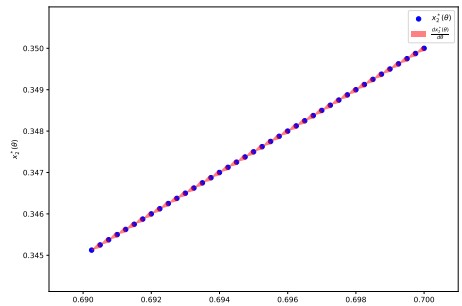

(a) 40 iterations of gradient descent for minimizing $x_1^\star(\theta)$ for the problem equation 32.

(b) 40 iterations of gradient descent for minimizing $x_2^\star(\theta)$ for the problem equation 32.

**C.1.3.2   Infeasible case ($\theta > 2$).**   For any $\theta > 2$, the LP is infeasible. The corresponding ECJs are the least-square solutions to

$$
\arg\min_{b_1, b_2, b_3, b_4} \left\| \begin{bmatrix} 0 & C^\top & 0 & 0 \\ C & 0 & (I - \Pi_1) & 0 \\ 0 & -I & -\Pi_1\Pi_2 & (1 - \Pi_2)C \end{bmatrix} \begin{bmatrix} b_1 \\ b_2 \\ b_3 \\ b_4 \end{bmatrix} + \begin{bmatrix} \frac{\delta \mathcal{L}_i}{\delta x} \\ 0 \\ 0 \end{bmatrix} \right\|_2^2 \text{ for } i \in \{1, 2\},
$$

with $P_1 = \begin{bmatrix} 0 & 0 & 0 & 0 & 0 \\ 0 & 0 & 0 & 0 & 0 \\ 0 & 0 & 1 & 0 & 0 \\ 0 & 0 & 0 & 0 & 0 \\ 0 & 0 & 0 & 0 & 1 \end{bmatrix}$ and $P_2 = \begin{bmatrix} 1 & 0 & 0 & 0 & 0 \\ 0 & 1 & 0 & 0 & 0 \\ 0 & 0 & 0 & 0 & 0 \\ 0 & 0 & 0 & 1 & 0 \\ 0 & 0 & 0 & 0 & 0 \end{bmatrix}$. The linear system within the $\ell_2$ norm is feasible and QPLayer outputs as ECJs $\frac{\partial \mathcal{L}_1}{\partial \theta} = \frac{\partial \mathcal{L}_2}{\partial \theta} = \frac{1}{3}$. Figure 9a and Figure 9b show that following such directions allows using gradient descent for minimizing $\mathcal{L}_1$ and $\mathcal{L}_2$.

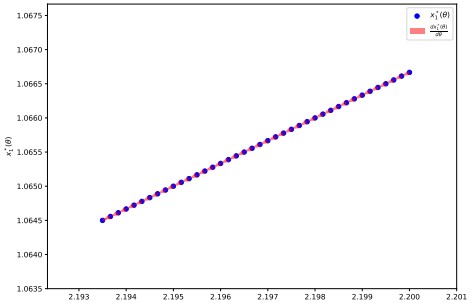 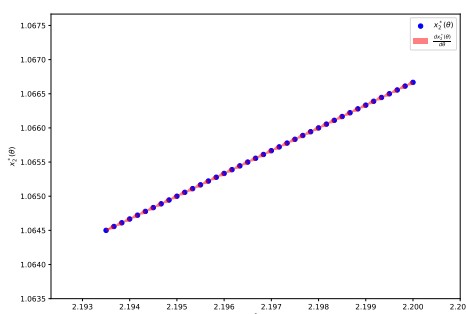

(a) 40 steps of gradient descent applied to minimize $x_1^\star(\theta)$ starting from $\theta_0 = 2.2$, when considering the infeasible LP defined by equation 32.

(b) 40 steps of gradient descent applied to minimize $x_2^\star(\theta)$ starting from $\theta_0 = 2.2$, when considering the infeasible LP defined by equation 32.

## C.2 TIMING BENCHMARKS

In this section, we report our numerical results and compare them against state-of-the-art frameworks on a set of standard experiments. We outline detailed timings for differentiating solutions on a set of different QPs. First, Table 2 shows the results for a few randomly generated QPs. Second, Table 3 reports the average time spent in the differentiation procedure on four different learning tasks. Additional experiments are provided in Appendix C.2.

In the first set of experiments (see Appendix C.2.1), QPLayer is compared to OptNet, CvxpyLayer, JaxOpt, and Alt-Diff. For all the other experiments, QPLayer is benchmarked only against approaches available within the PyTorch framework (i.e., OptNet and CvxpyLayer[4]).

### C.2.1 RANDOM QPS

In this first set of experiments, we generated random QPs with 100 variables, 50 equality, and 50 inequality constraints. We solve and backpropagate through all those QPs for different forward pass accuracies. We then average the results over 5 trails and report the timings in the spirit of (Sun et al., 2022).

For a batch size of 100, Table 2 shows that QPLayer is almost 4 times faster than OptNet (the second fastest approach). We can see similar performance for a batch size of 1 (see Table 4, QPLayer is about 4 times faster than OptNet (the second fastest approach) for all target accuracies). We observe that the speed gain is mostly due to the forward pass speed-up, enabled by the use of ProxQP and thread parallelization. It is also confirmed by Table 5, which reproduces in Table 5 the serial forward timing benchmark proposed in (Amos & Kolter, 2017, Section 4.1). It exhibits from 5 to 9 times faster computation times.

### C.2.2 LEARNING TASKS

For this second set of experiments, we report the numerical results obtained on 4 traditional learning tasks (namely MNIST classification, signal denoising, Sudoku solving and cart-pole experiment). For all experiments, we report the average (over all epochs) time spent in the forward and backward passes. Table 3 reports that QPLayer is 3 to 10 times faster than the second fastest approach (i.e., 3 times faster on the classification task, 4 times faster on the Sudoku, about 10 times faster for the denoising and 7 times faster for the cart-pole experiments). In all cases, the test loss incurred using QPLayer is either similar (for the denoising and cart-pole tasks) or far better than its competitor layers (about 2 to 3 times smaller for the classification and Sudoku experiments).

More precisely, the first three experiments reproduce the ones originally described in (Amos & Kolter, 2017, Sections 4.2 to 4.4). A complete description of the cart-pole swing-up task is detailed

---

[4]Alt-Diff exhibited too slow performances for a fair and reasonable comparison. Note that (Sun et al., 2022) have not yet proposed an Alt-Diff layer deriving all QP Jacobians. Therefore, we have included in our benchmark an open-source implementation of Alt-Diff based on their work.

in Appendix D.2. These tasks involve learning convex feasible QPs using Adam optimizer (Kingma & Ba, 2014). The first three experiments are run with the default batch sizes and the number of epochs fixed by the original authors (i.e., batch size equals 64 for classification, 150 for denoising and Sudoku tasks; 30 epochs for classification, and 20 for denoising and Sudoku tasks). We run the cart-pole example with batch size 1 for 800 epochs. For the first three experiments, we use the same QP layer models as (Amos & Kolter, 2017), except that we have changed the backends for executing the forward and backward passes (using either Qpth, QPLayer[5] or CvxpyLayer). Finally, we have used the following learning rates for running the benchmarks: $10^{-3}$ for classification, $10^{-5}$ for denoising, $5 \times 10^{-2}$ for Sudoku, and $10^{-1}$ for cart-pole tasks.

| Forward tolerance $\epsilon$ | $10^{-1}$ | $10^{-2}$ | $10^{-3}$ |
|---|---|---|---|
| Forward (ms) | **72.43** ($\pm7.28$) | **71.89** ($\pm3.0$) | **72.12** ($\pm4.21$) |
| Backward (ms) | 18.14 ($\pm1.01$) | 18.03 ($\pm0.17$) | 18.12 ($\pm0.37$) |
| **QPLayer total (ms)** | **90.57** | **89.92** | **90.24** |
| Forward (ms) | 340.62 ($\pm7.06$) | 348.68 ($\pm3.51$) | 349.44 ($\pm2.44$) |
| Backward (ms) | **6.39** ($\pm0.16$) | **6.61** ($\pm0.41$) | **6.66** ($\pm0.38$) |
| **OptNet total (ms)** | 347.01 | 355.29 | 356.10 |
| Forward (ms) | 123.35 ($\pm13.70$) | 196.75 ($\pm38.13$) | 281.54 ($\pm64.18$) |
| Backward (ms) | 546.91 ($\pm55.58$) | 622.45 ($\pm66.63$) | 723.34 ($\pm66.26$) |
| **JaxOpt total (ms)** | 670.16 | 819.20 | 1004.88 |
| Forward (ms) | $1.16(\pm0.038)\times10^3$ | $1.19 (\pm0.015)\times10^3$ | $1.24 (\pm0.017)\times10^3$ |
| Backward (ms) | 187.52($\pm8.59$) | 187.74 ($\pm11.54$) | 197.18 ($\pm7.99$) |
| **CvxpyLayer total (ms)** | $1.35\times10^3$ | $1.38\times10^3$ | $1.43\times10^3$ |
| Forward (ms) | Time limit | Time limit | Time limit |
| Backward (ms) | Time limit | Time limit | Time limit |
| **Alt-Diff total (ms)** | Time limit | Time limit | Time limit |

Table 2: Timings for deriving all Jacobians of random QPs with different forward pass accuracies and batch size 100.

---

[5]Yet two differences should be noted: QPLayer learns LP for the Sudoku experiment. We have not imposed zero Hessian for the CvxpyLayer even if it could learn it, as it would display worse results. Furthermore, for the denoising experiment, QPLayer learns the lower and upper bounds simultaneously.

| Learning Tasks | QPLayer | OptNet | CvxpyLayer |
|---|---|---|---|
| **cart-pole** | | | |
| Forward (ms) | **55.20** ($\pm$6.93) | 615.84 ($\pm$16.15) | 629.70 ($\pm$30.42) |
| Backward (ms) | **39.27** ($\pm$2.17) | 61.26 ($\pm$2.84) | 101.88 ($\pm$1.11) |
| Final test loss | **0.02556** | 0.02604 | 0.02566 |
| **Sudoku** | | | |
| Forward (ms) | **87.39** ($\pm$16.69) | 454.48 ($\pm$22.22) | 772.77 ($\pm$33.89) |
| Backward (ms) | 20.80 ($\pm$1.70) | **8.07** ($\pm$0.42) | 192.99 ($\pm$9.81) |
| Final test loss | **6.19**$\times10^{-4}$ | 0.0017 | 0.389 |
| **denoising** | | | |
| Forward (ms) | **247.89** ($\pm$17.03) | 2827.48 ($\pm$618.78) | Error |
| Backward (ms) | 52.99 ($\pm$2.75) | **40.57** ($\pm$2.98) | Error |
| Final test loss | **3281.84** | 3529.2029 | Error |
| **classification** | | | |
| Forward (ms) | **26.77**($\pm$3.63) | 102.62 ($\pm$18.23) | Error |
| Backward (ms) | **11.12** ($\pm$3.01) | 16.58 ($\pm$12.05) | Error |
| Final test loss | **0.1363** | 0.3264 | Error |

Table 3: Timings and final loss of 4 learning tasks. CvxpyLayer errors arise due to failures in filling the disciplined parametrized programming (DPP) form of the quadratic cost.

| Forward tolerance $\epsilon$ | $10^{-1}$ | $10^{-2}$ | $10^{-3}$ |
|---|---|---|---|
| Forward (ms) | **1.30** ($\pm$0.17) | **1.42** ($\pm$0.19) | **1.55** ($\pm$0.20) |
| Backward (ms) | 0.77 ($\pm$0.01) | 0.70 ($\pm$0.02) | 0.71 ($\pm$0.02) |
| **QPLayer total (ms)** | **2.07** | **2.12** | **2.26** |
| Forward (ms) | 6.93 ($\pm$0.82) | 6.85 ($\pm$0.05) | 7.49 ($\pm$0.04) |
| Backward (ms) | **0.75** ($\pm$12) | **0.70** ($\pm$0.01) | **0.70** ($\pm$0.01) |
| **OptNet total (ms)** | 7.68 | 7.55 | 8.19 |
| Forward (ms) | 7.99 ($\pm$0.93) | 12.82 ($\pm$2.67) | 18.82 ($\pm$4.31) |
| Backward (ms) | 24.71 ($\pm$2.53) | 30.47 ($\pm$2.71) | 36.43 ($\pm$3.93) |
| **JaxOpt total (ms)** | 32.70 | 43.29 | 55.25 |
| Forward (ms) | 411.20 ($\pm$3.90) | 415.94 ($\pm$2.63) | 422.28 ($\pm$2.65) |
| Backward (ms) | 6.13 ($\pm$0.02) | 6.34 ($\pm$0.33) | 6.26 ($\pm$0.06) |
| **CvxpyLayer total (ms)** | 417.33 | 422.24 | 428.54 |
| Forward (ms) | 6.00 ($\pm$0.85)$\times10^3$ | 38.66 ($\pm$7.11)$\times10^3$ | 114.20 ($\pm$31.48)$\times10^3$ |
| Backward (ms) | 0.99 ($\pm$0.21) | 0.98 ($\pm$0.13) | 1.05 ($\pm$0.14) |
| **Alt-Diff total (ms)** | 6.00$\times10^3$ | 36.66$\times10^3$ | 114.20$\times10^3$ |

Table 4: Averaged timings for the forward and backward passes when computing all Jacobians of randomly generated feasible QPs (with 100 primal variables, 50 equality constraints and 50 inequality constraints) considering different forward pass accuracies. The batch size is 1. QPLayer has the best total timings for all accuracies.

| QP and Batch sizes | QPLayer (ms) | OptNet (ms) | CvxpyLayer (ms) |
|---|---|---|---|
| Batch $= 1$, $n = 100$, $n_e = 0$, $n_i = 50$ | **0.88** ($\pm 0.06$) | 8.01 ($\pm 0.89$) | 334.06 ($\pm 7.83$) |
| Batch $= 1$, $n = 100$, $n_e = 50$, $n_i = 50$ | **1.09** ($\pm 0.10$) | 8.35 ($\pm 0.67$) | 406.18 ($\pm 1.44$) |
| Batch $= 1$, $n = 100$, $n_e = 0$, $n_i = 100$ | **1.22** ($\pm 0.08$) | 7.62 ($\pm 0.32$) | 422.59 ($\pm 4.34$) |
| Batch $= 1$, $n = 100$, $n_e = 50$, $n_i = 100$ | **1.75** ($\pm 0.17$) | 13.95 ($\pm 2.03$) | 522.80 ($\pm 3.96$) |
| Batch $= 1$, $n = 100$, $n_e = 100$, $n_i = 50$ | **1.26** ($\pm 0.14$) | 16.77 ($\pm 3.41$) | 521.18 ($\pm 16.15$) |
| Batch $= 1$, $n = 100$, $n_e = 100$, $n_i = 100$ | **1.45** ($\pm 0.28$) | 18.33 ($\pm 5.35$) | 621.25 ($\pm 9.67$) |
| Batch $= 64$, $n = 100$, $n_e = 0$, $n_i = 50$ | **30.16** ($\pm 5.51$) | 173.95 ($\pm 12.82$) | 771.49 ($\pm 59.74$) |
| Batch $= 64$, $n = 100$, $n_e = 50$, $n_i = 50$ | **48.60** ($\pm 9.19$) | 183.84 ($\pm 9.99$) | 818.75 ($\pm 5.42$) |
| Batch $= 64$, $n = 100$, $n_e = 0$, $n_i = 100$ | **44.68** ($\pm 5.95$) | 176.88 ($\pm 2.30$) | 820.85 ($\pm 12.38$) |
| Batch $= 64$, $n = 100$, $n_e = 50$, $n_i = 100$ | **55.77** ($\pm 6.38$) | 243.15 ($\pm 27.86$) | 1126.99 ($\pm 34.33$) |
| Batch $= 64$, $n = 100$, $n_e = 100$, $n_i = 50$ | **51.66** ($\pm 8.51$) | 231.11 ($\pm 13.75$) | 1177.89 ($\pm 132.99$) |
| Batch $= 64$, $n = 100$, $n_e = 100$, $n_i = 100$ | **62.52** ($\pm 7.84$) | 276.18 ($\pm 31.29$) | 1314.47 ($\pm 69.60$) |
| Batch $= 128$, $n = 100$, $n_e = 0$, $n_i = 50$ | **62.94** ($\pm 9.52$) | 620.91 ($\pm 3.52$) | 1202.61 ($\pm 88.32$) |
| Batch $= 128$, $n = 100$, $n_e = 50$, $n_i = 50$ | **78.01** ($\pm 4.06$) | 667.55 ($\pm 5.86$) | 1295.50 ($\pm 24.63$) |
| Batch $= 128$, $n = 100$, $n_e = 0$, $n_i = 100$ | **89.48** ($\pm 7.09$) | 653.32 ($\pm 5.54$) | 1267.83 ($\pm 30.85$) |
| Batch $= 128$, $n = 100$, $n_e = 50$, $n_i = 100$ | **111.01** ($\pm 8.21$) | 774.08 ($\pm 20.90$) | 1739.41 ($\pm 70.53$) |
| Batch $= 128$, $n = 100$, $n_e = 100$, $n_i = 50$ | **96.23** ($\pm 9.47$) | 811.44 ($\pm 25.29$) | 1896.52 ($\pm 284.89$) |
| Batch $= 128$, $n = 100$, $n_e = 100$, $n_i = 100$ | **119.13** ($\pm 10.17$) | 934.26 ($\pm 87.79$) | 2059.44 ($\pm 147.77$) |

Table 5: Timing benchmarks of different backends used for solving a forward pass for different batch sizes. $n_e$ stands for the number of equality constraints.

## C.3 TRAINING WITH LARGE LEARNING RATES

In this section, we assess the numerical robustness of QPLayer on traditional learning tasks by demonstrating that it can be trained with larger learning rates than other approaches, potentially resulting in improved attraction pools.

More precisely, we ran the MNIST classification and denoising tasks described in **??** with SGD and larger learning rates and reported the corresponding results. We measured the final test loss and error reached after 30 epochs and the standard deviation over the last 10 epochs of the test loss and test error. The results are averaged over 10 seeds and the experiment is performed for different learning rates.

As described in Appendix C.2.2, for those tasks, the QP layers need to learn all the model parameters (i.e., $H$, $g$, $C$, and $u$). We observe that it generates potentially very ill-conditioned problems when the forward or the backward passes are not solved accurately enough. This phenomenon appears to be amplified with larger learning rates. In those situations, it appears that robust solution methods (e.g., allowing for temporary infeasible, or ill-conditioned problems) are critical.

Figure 10a and Figure 10b show that for too high learning rates (i.e., $10^{-4}$ or $10^{-5}$ for denoising task and $10^{-2}$ for the classification task) the OptNet layer generate errors, whereas it is never the case for QPLayer. Furthermore, for low learning rate levels (i.e., $10^{-6}$ or $10^{-7}$ for the denoising task and $10^{-3}$ and $10^{-4}$ for the classification task), the final loss reached is similar but with a less important noise amplitude level when using QPLayer ( Figure 11 provides robustness statistics of the classification task using the prediction error rate of the two layers). Finally, QPLayer is capable of being trained with a larger learning rate (i.e., $10^{-4}$ for the denoising task and $10^{-2}$ for the classification task). Also,

note that CvxpyLayer fails to be run in all these robustness experiments because the Hessian part of the quadratic model fails to fit the required DPP form (Amos et al., 2018, Section 4.1).

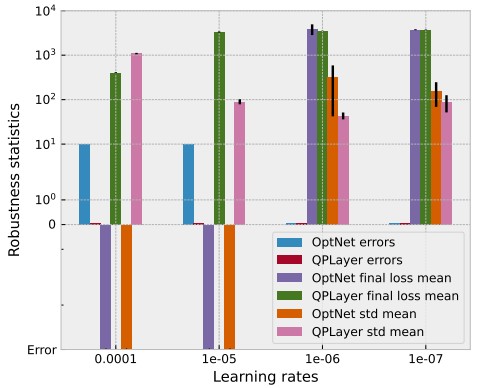

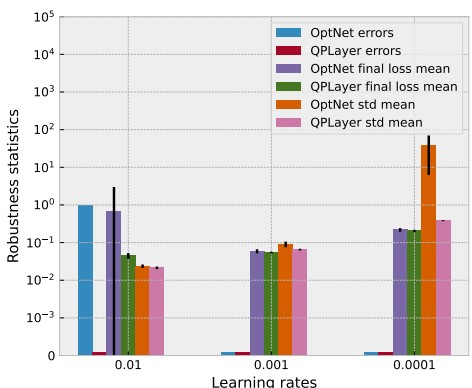

(a) Robustness statistics for denoising task.

(b) Robustness statistics for MNIST classification task.

Figure 10: Robustness statistics for denoising and MNIST classification tasks: number of errors (i.e., NaNs), averaged final loss reached after 30 epochs (with 95% confidence intervals), and averaged standard deviations over the last 10 epochs (with 95% confidence intervals). Results are averaged over 10 seeds. CvxpyLayer fails to be trained for all these tasks.

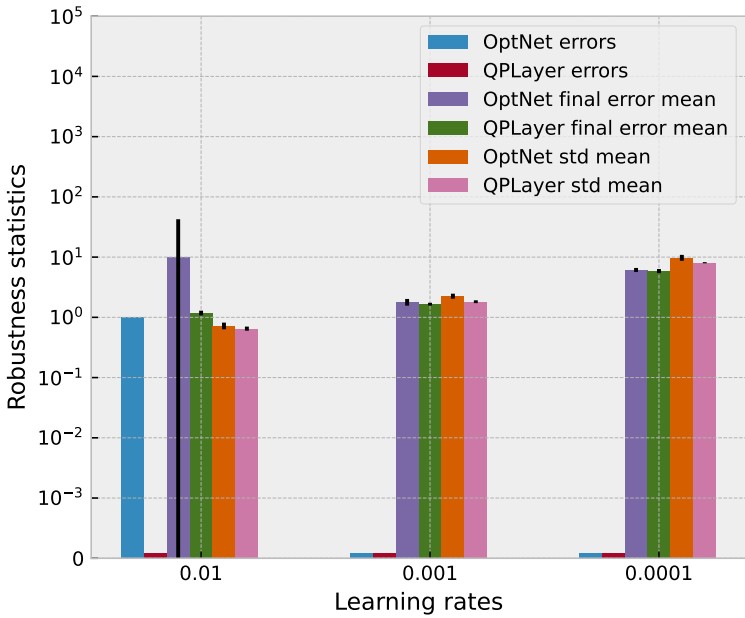

Figure 11: Robustness statistics for the MNIST classification: number of errors (i.e., NaNs), averaged final prediction error reached after 30 epochs (with 95% confidence intervals), and averaged standard deviations over the last 10 epochs (with 95% confidence intervals). Results are averaged over 10 seeds.

**Remark 3** (Numerical differences with OptNet). *Our approach offers a few numerical advantages compared to (Amos & Kolter, 2017). In particular, a numerical matrix factorization is at the center of most popular techniques for differentiating through QPs. This factorization procedure represents one of the main bottlenecks in the computational costs. In our approach, we need to factorize smaller and better-conditioned symmetric matrices.*

*The formulation exploited by OptNet consists of a larger linear system to compute its Jacobians. More precisely, they factorize a matrix of the form:*

$$K = \begin{bmatrix} H & 0 & C^\top \\ 0 & D(z^\star) & D(t^\star) \\ C & I & 0 \end{bmatrix},$$

*where $t^\star = Cx^\star - u$ and $D(z^\star)$ corresponds to a diagonal matrix whose diagonal entries correspond to $z^\star$. For obtaining a symmetrized version that can be factorized with efficient methods, the second row block is scaled by $D(1/t^\star)$ (Amos & Kolter, 2017, Section 3.1). Yet, symmetrization comes at the price of being more sensitive to the localization of the solution w.r.t the constraints. Indeed, if $x^\star$ lies on the boundary, i.e., $C_I x^\star - u_I = 0$ for some component index $I$, the conditioning of the matrix is degraded, as OptNet needs to divide by zeros (or small clamped numbers in practice).*

*On our side, the formulation for feasible QPs relies on a smaller matrix, which is symmetric and better-conditioned (it does not require scaling rows by values that are potentially zeros, see equation 10).*

## D   EXPERIMENTAL SETUP

This section details the optimization architectures used for the Sudoku tasks described in Section 4.1. The cart-pole task mentioned in Appendix C.2.2 is detailed in Appendix D.2.

### D.1   LAYER ARCHITECTURE FOR THE SUDOKU PROBLEM

The layer architecture used by OptNet Amos & Kolter (2017) for the Sudoku problem is described in Figure 12. In contrast, the QPLayer architecture (which does not require structural feasibility of the QPs) is described in Figure 13. The QPLayer architecture allows learning more structured constraints, such as the Sudoku constraint "$Ax = 1$", which cannot be done, as is, with OptNet (which requires the QPs to be structurally feasible).

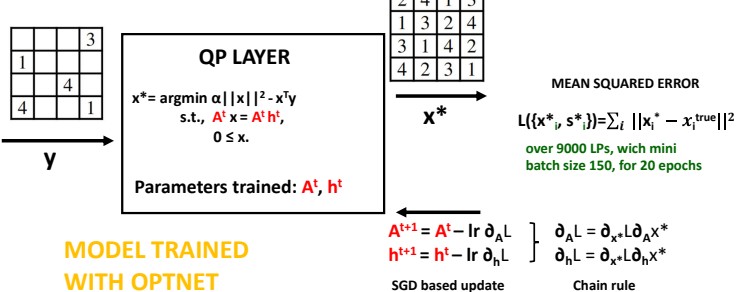

Figure 12: Strictly convex QP layer trained using OptNet in Section 4.1.1 (as in Amos & Kolter (2017)). The constraint matrix and an extra variable $z_0$ are learned in order to be sure that the QP is always primal feasible (structural feasibility).

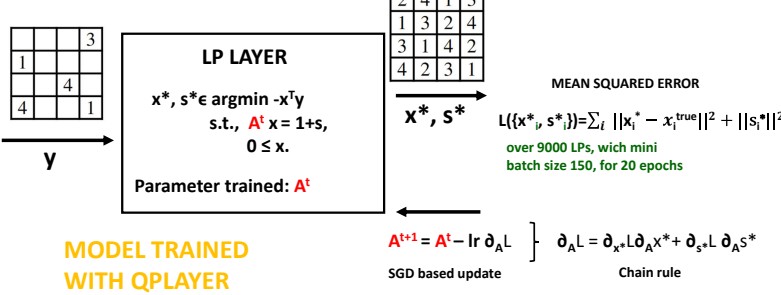

Figure 13: An LP layer trained as allowed by QPLayer in Section 4.1.2. It enables for more flexibility in the problem to be learned (only the constraint matrix $A$ is learned). The optimal shift $s^*$ is a new output variable minimized in the loss, in order to learn at test time a feasible LP layer.

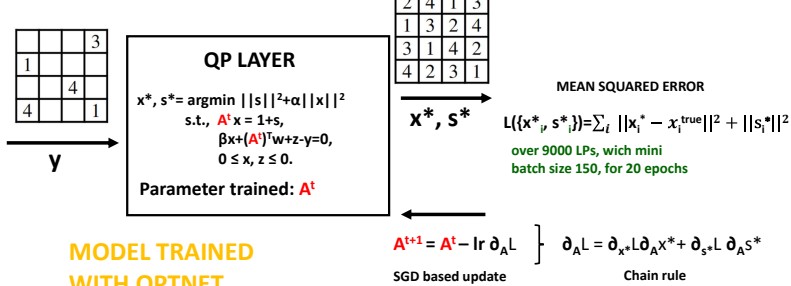

Figure 14: A QP layer formulating equation QP-H($\theta$) as a convex QP problem. The optimal shift $s^*$ is a new output variable minimized in the loss, in order to learn at test time a feasible QP layer. The resulting forward pass considers a larger and potentially harder problem to solve. In Section 4.1.2, this layer is trained using OptNet after adding a small strictly convex quadratic (i.e., $5 \times 10^{-5} \| \begin{bmatrix} x \\ y \\ z \\ s \end{bmatrix} \|_2^2$).

### D.2 DESCRIPTION OF THE CART-POLE PROBLEM

The cart-pole system (Anderson, 1989) is a classic control problem used for benchmarking control algorithms. The system we consider consists of an extended model with dry friction on the joints of the cart-pole, namely on the prismatic cart joint and the revolute joint of the pole. It makes the dynamics non-smooth. It is described by a set of differential equations relating the position, velocity, acceleration, angle, and angular velocity of the cart and pole plus the additional friction forces. The static friction forces on each joint can be obtained by solving a QP problem, see equation 33 and (Le Lidec et al., 2021).

**Task:** The initial position of the cart-pole system consists of the pole hanging down vertically. The objective of the task is to move the cart in such a way as to swing the pole up and keep it balanced in the upright position. To swing the pole up, the control inputs may involve moving the cart back and forth in a particular pattern that generates the necessary forces to overcome the friction and accelerate the pole in the desired direction.

The forward dynamics with friction

$$Ma = \tau + \lambda,$$

can be re-written in terms of velocity and impulses with timestep $\Delta t$ as

$$v = v_f + M^{-1}\lambda\Delta t = v_f + M^{-1}\Lambda,$$

were $M$ is the inertia matrix of the system, $a \in \mathbb{R}^{n_v}$ is the joint acceleration, $\tau \in \mathbb{R}^{n_v}$ the joint torque, $v_f$ the free velocity of the system without friction and $\lambda \in \mathbb{R}^{n_v}$ the dry friction force on every joint. To obtain the friction impulse $\Lambda$ corresponding to the friction coefficient $\eta$, the following quadratic problem can be solved:

$$\min_{\Lambda} \frac{1}{2}\Lambda^T M^{-1}\Lambda + v_f^T \Lambda \tag{33}$$
$$\text{s.t. } |\Lambda| \leqslant \eta.$$

Its Lagrangian $L$ can be written as follows

$$L(\Lambda, y) := \frac{1}{2}\Lambda^T M^{-1}\Lambda + v_f^T \Lambda + y^T(|\Lambda| - \eta),$$

which leads to the KKT system:

$$M^{-1}\Lambda + v_f + \text{diag}(\text{sign}(\Lambda))y = 0, \tag{34}$$
$$|\Lambda| \leqslant \eta, \tag{35}$$
$$y \geqslant 0, \tag{36}$$
$$y \odot [|\Lambda| - \eta] = 0, \tag{37}$$

where $\odot$ stands for the standard Hadamard product. Considering the case where the friction force is within the friction cone for a specific joint $j$, i.e., $|\Lambda_j| < \eta_j$, the joint is then not moving. We see from equation 37 that $y_j = 0$ satisfies equation 36 and we get from equation 34

$$M^{-1}\Lambda_j = -v_{f,j}.$$

Consequently, the friction impulse is acting in the opposite direction than the joint torque $\tau_j$ and with a magnitude that is canceling out the free velocity. If $|\Lambda_j| = \eta_j$, the joint will no longer be blocked by the friction forces and will thus start moving. We see from equation 34

$$M^{-1}\Lambda_j + v_{f,j} = v_j = -\text{diag}(\text{sign}(\Lambda_j))y_j. \tag{38}$$

As $y \geqslant 0$, equation 38 shows that $\Lambda$ is opposed to the velocity of the joint.

Optimal control algorithms such as Differential Dynamic Programming (DDP) can be used to compute the optimal trajectory that minimizes a cost function over a finite time horizon, subject to the dynamics of the cart-pole system and control input constraints.

These methods take advantage of the derivatives of the dynamics to efficiently control physical systems. In the presence of non-smooth dynamics, such a class of algorithms is likely to fail due, for instance, to the presence of discontinuities in the dynamics derivatives or because of the non-informative gradient (Lidec et al., 2022). In the cart-pole benchmark, randomized smoothing, as proposed by (Lidec et al., 2022; Suh et al., 2021) is used to cope with the non-smooth dynamical system. For the optimal swing-up trajectory with 20 timesteps, 5 random samples with a uniform Gaussian noise are generated. The random noise is applied to the input controls, and the dynamics are calculated for each of them and afterward averaged in the forward pass, resulting in informative gradients in the backward pass. equation 33, has to be solved for every random sample in every timestep.

