# OpenReview forum: "Leveraging augmented-Lagrangian techniques for differentiating over infeasible quadratic programs in machine learning"
_ICLR.cc/2024/Conference — ICLR 2024 spotlight_

### Official Review · Reviewer_uDj8 · 2023-10-31

**Soundness:** 3 good
**Presentation:** 3 good
**Contribution:** 2 fair
**Rating:** 6
**Confidence:** 4

**Summary:**

The paper studies the problem of differentiating through quadratic programming layers. These are layers whose output is the solution to some quadratic programming problem, whose problem data depends on the input to the layer.

The main contribution of the paper is to provide a more efficient way to compute the Jacobian of the solution mapping which maps inputs to the layer to the solution of the quadratic programming problem. Besides this, there are also a number of experiments demonstrating the efficacy of this proposed method on the sudoku problem, comparing to alternative methods like optnet and cvxpylayer.

A major obstacle in quadratic programming layers has been maintaining feasibility of the quadratic program; previous works have always parametrized or constrained the layers in a way to ensure feasibility but this limits expressiveness. A key advantage of the method proposed by the paper is that it can handle both feasible and infeasible quadratic programs by extending the definition of a solution to account for infeasibility. In the case of infeasibility, the solution is considered to be a solution of the closest feasible quadratic program in a least-squares sense. This is accomplished using an augmented Lagrangian approach to representing the quadratic program that is more flexible and allows to better treat this infeasibility, at least empirically.

The paper also introduces a notion of extended conservatve Jacobian as a way to make sense of the nonsmooth "Jacobian" like objects computed by automatic differentiation in the case of infeasibility. There are some results showing that, under very strict hypotheses, the extended conservative Jacobians correspond to ordinary conservative Jacobians or ordinary Jacobians.

Small comment: in section 3.1 "hierarchic" is not a word.

**Strengths:**

The paper is very rigorous and mathematically precise in its statements and proofs. Furthermore, the method is very practical in the sense that it allows for a much broader class of quadratic programming layers than previous works did while at the same time offering computational advantages in terms of accuracy, time for the forward pass, and total time for the forward+backward passes. These advantages are shown to hold for many different problem settings, not only the sudoku problem but also the cart-pole problem (optimal control), and denoising and classification tasks coming from computer vision. The experiments are very comprehensive and convincing in terms of the empirical performance of this method vs prior works on quadratic programming layers.

**Weaknesses:**

The theoretical claims in the paper are correct but they are basic and unsatisfying. While it's shown that G, a sort of KKT mapping, is path differentiable with respect to its arguments x,z, and t, this is insufficient for rigorously differentiating through the quadratic program layer. It must also be shown that the solution mappings themselves, x\*, z\*, and t\* are path differentiable with respect to the theta. Otherwise, knowing that G alone is path differentiable is not useful. The path differentiability of x\*, z\*, and t\* is rigorously proved in Lemma 3 but the assumptions of this lemma are so strong that it is no longer applicable to quadratic programming layers. It requires that the objective function in the quadratic program is no longer quadratic (indeed, it must be linear in the lemma) and that it does not depend on theta anymore (g is constant).

This does not appear to be a trivial problem; proving that the solution maps are path differentiable will require assumptions that negate some of the proposed contributions of the paper, i.e., we will no longer be able to treat such a broad class of problems and this might even exclude infeasible problems, which is a supposed motivation of this work. To be fair, this is a problem of other frameworks as well (as the authors themselves point out) and this paper should not be singled out for this shortcoming, but it is indeed a **major** shortcoming in terms of the theoretical contributions of this work.

Because of this drawback with respect to the theory, I find the claims that this method handles infeasible problems a bit unsatisfactory - it seem to be based entirely on the experimental observations that it works but there is no rigorous justification that I can find.

**Questions:**

Is there any hope to handle the gaps that you outline in 3.4.3? Even for feasible programs, it's not always guaranteed that x\* is a function since there can be many solutions, so how would it be possible to pursue path differentiability?

Since the augmented Lagrangian approach for possibly infeasible QPs already existed, is the contribution here just to combine it with deep learning?

---

> ### Author Response · Authors · 2023-11-16
>
> We first would like to thank the reviewer for the time spent evaluating our work and for the constructive feedback.
>
> ## Questions
>
> **Is there any hope to handle the gaps that you outline in 3.4.3? Even for feasible programs, it's not always guaranteed that x is a function since there can be many solutions, so how would it be possible to pursue path differentiability?**
>
> We thank the reviewer for this very relevant remark. As explained in the original work about path differentiability [1], this notion has been introduced to help explain better the behavior of Stochastic Gradient Descent-Based algorithms applied to nonsmooth functions and Deep Learning frameworks. Yet, as pointed out by the reviewer, there are indeed still unsatisfactory mathematical gaps. From what we can tell, one major unresolved gap is the scope of applicability of the Implicit function Theorem (IFT) (including all its extensions, for example when dealing with Conservative Jacobians). Most of the time, the IFT requires at some point nonsingular matrices, and hence applies only under restrictive assumptions. In particular, they do not generically hold during the training.
>
> Nevertheless, we see at least one hope. There are known examples (see e.g., the discussion in [2, Section 5.4]) when the IFT can still be applied (requiring judicious tricks) even when involved with "singular" matrices. Hopefully, these techniques could be one day formalized to extend this theorem to more generic instances. These thoughts have motivated us to define the notion of "Extended Conservative Jacobian" via a least-square relaxation (since least-square solutions include the cases when the underlying linear system has multiple solutions). Our "positive" experimental results could hence probably be interpreted by the fact that the scope of the IFT is currently underestimated.
>
> We will add this reference and this discussion in section 3.4.3.
>
> **Since the augmented Lagrangian approach for possibly infeasible QPs already existed, is the contribution here just to combine it with deep learning?**
>
> The primary objective of this paper is to extend the scope of trainable convex QP layers, in order to be able to capture more domain-specific knowledge or priors. One such way is considering training layers that are not structurally feasible. Our work presents theory and practical algorithms for training more natural types of layers, that were not common before, due to our limited technical capabilities.
>
> In this context, the AL-based algorithms must be seen more as a practical tool for solving QP-H. It proves to be extremely profitable in our case (see e.g., the experiments conducted in Section 4.1.2 where QP-H is solved via a QP reformulation using OptNet).
>
> ## Weaknesses
>
> ***The path differentiability of x, z, and t is rigorously proved in Lemma 3 but the assumptions of this lemma are so strong that it is no longer applicable to quadratic programming layers. It requires that the objective function in the quadratic program is no longer quadratic (indeed, it must be linear in the lemma) and that it does not depend on theta anymore (g is constant).***
>
> We agree with the reviewer that the scope of applicability of this lemma is a bit restrictive. Yet, we also would like to point out that the result is not that trivial. Indeed, it is quite lucky that a solution of an infeasible Linear Program is path differentiable (i.e., its Jacobian has a chain rule).
>
> We plan, in our next works to investigate more elaborate extensions of this lemma (in more generic conic cases, including the complete quadratic case). We focused more precisely on the LP setting for this lemma since it has a close connection with the Sudoku experiment that follows.
>
> ## REFERENCES
>
> [1] Jérôme Bolte and Edouard Pauwels. Conservative set valued fields, automatic differentiation,
> stochastic gradient method and deep learning
>
> [2] Krantz, S. G. and Parks, H. R. (2002). The implicit function theorem: history, theory, and applications. Springer Science & Business Media

---

> > ### Comment · Reviewer_uDj8 · 2023-11-21
> >
> > Thank you for your response. I still see shortcomings in the theoretical contributions of the paper (in particular, for layers with parametrized quadratic objectives) but I agree the empirical findings are very strong. For these reasons I maintain my original score.

---

### Official Review · Reviewer_vnSN · 2023-11-01

**Soundness:** 3 good
**Presentation:** 2 fair
**Contribution:** 3 good
**Rating:** 8
**Confidence:** 3

**Summary:**

This paper considers the problem of training neural network with quadratic program (QP) layers. In order to differentiate the quadratic program, previous work has to enforce the feasibility by ignoring some constraints. In contrast, this work improves the QP layer and allows directly differentiating through infeasible QP layers. The this end, the authors introduce a slack variable (which measures the "degree" of infeasibility) and minimize the the Euclidean of the slack variable in a hierarchical quadratic program. To differentiate through the hierarchical, the author propose using the extended conservative Jacobian.

They show the improve QP layer enables learning linear programs and quadratic programs without neglecting any constraints. In addition, the proposed method stabilizes training and leads to better performance.

**Strengths:**

- The idea is well-motivated and experiments are convincing. This paper solves an important problem in learning QP layers. The proposed method could greatly improve the robustness of QP layer training and handle all QP layers (including the infeasible ones) in a principle manner.
- The technique in this paper could be potentially generalized to all differentiable convex layers.

**Weaknesses:**

- This paper is very technical and the presentation should be improved. For example, the non-linear map (G) is introduced without any background information. The authors should add more text to motivate this non-linear map and explain how it is derived.
- Limited scope of applications. The experiment only shows training QP layers solving Sudoku problems. However, I am interested to see if the improved QP layer enables new applications.

**Questions:**

How is the hierarchical quadratic program (QP-H) solved numerically? Is it similar to phase-I method in interior point methods?

---

> ### Author Response · Authors · 2023-11-16
>
> We first would like to thank the reviewer for the time spent evaluating our work and for the constructive feedback.
>
> ## Questions
>
> ***How is the hierarchical quadratic program (QP-H) solved numerically? Is it similar to phase-I method in interior point methods?***
>
> We thank a lot the reviewer for this relevant question. We will add more details to amphisize more precisely this practical numerical aspect.
>
> QP-H is solved numerically through an Augmented Lagrangian (AL) algorithm (in practice we use the ProxQP solver). More precisely, when applied to a QP, an AL algorithm has the converging property of outputting a solution to QP-H, if the original QP is primally infeasible. We make use of this feature to solve QP-H. More details can be found in the following reference [1].
>
> ## Weaknesses
>
> ***This paper is very technical and the presentation should be improved. For example, the non-linear map (G) is introduced without any background information. The authors should add more text to motivate this non-linear map and explain how it is derived.***
>
> We thank a lot the reviewer for this feedback. We are going to highlight the key ideas of our work in order to improve the presentation and readability of the results.
>
> The map G is found via a change of variable from the KKT conditions for QP-H (detailed in equation 12 of the appendix, before the proof of Lemma 1). We will add this context to give more intuition about the origins of this map G.
>
> ***Limited scope of applications. The experiment only shows training QP layers solving Sudoku problems. However, I am interested to see if the improved QP layer enables new applications.***
>
> Again we thank a lot the reviewer for his feedback. The results of this work also apply to the classical examples where QPs layers need to be trained, in particular those in the appendix (e.g., cartpole, denoising and image classification). We will mention them more explicitly in the core of our work.
>
> We also plan to consider more practical applications involving infeasible QPs notably in the context of control-based and robotic experiments.
>
> ## References
>
> [1] Alice Chiche and Jean Charles Gilbert. How the augmented Lagrangian algorithm can deal with an infeasible convex quadratic optimization problem. Journal of Convex Analysis, 23(2), 2016.

---

> > ### Comment · Reviewer_vnSN · 2023-11-23
> >
> > I have read the authors' rebuttal and I remain positive about the paper.

---

### Official Review · Reviewer_7UKP · 2023-11-01

**Soundness:** 4 excellent
**Presentation:** 4 excellent
**Contribution:** 3 good
**Rating:** 8
**Confidence:** 3

**Summary:**

The paper proposes an approach for differentiating through quadratic programs (QP)s that might be primal infeasible. It provides all the necessary mathematical derivation and extensive numerical experiments including comparisons to state-of-the-art approaches like CvxpyLayers and OptNet. The paper provides a clear approach for this task (basically introducing primal slack variables and hence, considering the extended conservative Jacobian. Forward and backward mode autodiff rules are also provided.

**Strengths:**

Solving (constrained) QPs as a layer within a neural network can be a useful task. Hence, the topic of differentiating QPs that might be primal infeasible is very important. The mathematical derivation seems sound (though I did not fully check it), the presentation is very clear, and the experiments are also very convincing.

**Weaknesses:**

The approach behind CvxpyLayers seems to use a similar least-squares relaxation (as also stated in the paper). How does the presented approach differ from this?

**Questions:**

The approach behind CvxpyLayers seems to use a similar least-squares relaxation (as also stated in the paper). How does the presented approach differ from this?

---

> ### Author Response · Authors · 2023-11-16
>
> We first would like to thank the reviewer for the time spent evaluating our work and for the constructive feedback.
>
> ## Question
>
> ***The approach behind CvxpyLayers seems to use a similar least-squares relaxation (as also stated in the paper). How does the presented approach differ from this?***
>
> We thank a lot the reviewer for this relevant question.
>
> Fundamentally the least square relaxation is not applied to the same problem.
>
> In our work, we define the broader notion of the closest feasible QP solution and try differentiating through it via the Implicit Function Theorem (IFT). The least-square relaxation is then applied in order to output an "informative" Conservative Jacobian (named Extended Conservative Jacobian) when the IFT cannot be applied in this context. Note, that the least-square relaxation outputs also the IFT solution when it applies.
>
> CvxpyLayer considers differentiating via the IFT through a specific class of convex cones (named "Disciplined Parametrized Programs"). It applies directly the least-square relaxation to this specific problem.

---

### Official Review · Reviewer_sv98 · 2023-11-02

**Soundness:** 3 good
**Presentation:** 2 fair
**Contribution:** 3 good
**Rating:** 6
**Confidence:** 4

**Summary:**

The authors study the problem of calculating the gradient of optimal solutiosn to the input in the convex quadratic programming problem. One limitation of existing convex QP layers is that they assume the QP problem is always feasible during training. The main contribution of the paper is developing an augmented Lagrangian-based approch to calculate the gradient when the QP problem is not feasible.

**Strengths:**

1. The paper studies an important problem.
2. The paper proposes a new approach to calculate gradients in convex QP layers.

**Weaknesses:**

1. The paper is not easy to follow. See the comments below.

2. The analysis in the paper is not sufficient. See the comments below.

**Questions:**

1. The paper is not easy to follow. In some key sections or proofs, the authors suggests to read some other referecens to undertand the concepts. This makes reading the paper difficult for readers that are not working on this specific area of extended conservative Jacobian. To make the paper easier to read, the authors are suggested to add the basic concepts and ideas in the paper instead of requesting the audience to read a number of other reading materials.

2. To address the infeasibility issue for the QP layers, one straightfoward way is to introducing slack variables and add penalties about the constraint violation in the objective function. The authors are suggested to show the advantage of the proposed approach as compared to this straightforward approach.

3. One application of the QP layers is to use it as a building block for existing iterative algorithms (similar to the approach in [R1] and [R2]). The authors are suggested to analyze the convergence performance of the iterative algoritms when using the propsoed conservative Jacobian as a building block.


[R1] Donti, P.L., Rolnick, D. and Kolter, J.Z., 2021. DC3: A learning method for optimization with hard constraints. arXiv preprint arXiv:2104.12225.
[R2] Donti, P., Agarwal, A., Bedmutha, N.V., Pileggi, L. and Kolter, J.Z., 2021. Adversarially robust learning for security-constrained optimal power flow. Advances in Neural Information Processing Systems, 34, pp.28677-28689.

---

> ### Author Response · Authors · 2023-11-16
>
> We first would like to thank the reviewer for the time spent evaluating our work and for the constructive feedback.
>
> ## Main questions
>
> **The paper is not easy to follow. In some key sections or proofs, the authors suggests to read some other referecens to undertand the concepts. This makes reading the paper difficult for readers that are not working on this specific area of extended conservative Jacobian. To make the paper easier to read, the authors are suggested to add the basic concepts and ideas in the paper instead of requesting the audience to read a number of other reading materials.**
>
> We thank a lot the reviewer for his feedback. We will re-work the paper to make it easier to apprehend. Among others, we will add more basic concepts about Conservative Jacobians (CJ), notably, its definition, its original intuition, and some practical properties we use throughout our work.
>
> **To address the infeasibility issue for the QP layers, one straightfoward way is to introducing slack variables and add penalties about the constraint violation in the objective function. The authors are suggested to show the advantage of the proposed approach as compared to this straightforward approach.**
>
> We thank again the reviewer for this excellent question.
>
> Indeed, as pointed out by the reviewer, one straightforward way to address the infeasibility issue for the QP layers is to introduce slack variables and add penalties for the constraint violation in the objective function. It corresponds to a reformulation of QP-H($\theta$) as a convex QP.
>
> It turns out that when applied to a QP, an Augmented Lagrangian (AL) algorithm has the implicit converging property of outputting a solution to QP-H, if the original QP is primally infeasible. We make use of this feature to solve QP-H. More details can be found in the following reference [1].
>
> Regarding the comparison asked by the reviewer, we've already carried out similar tests (see section 4.1.2), and even implemented them in other frameworks. The results of the reformulation of QP-H as a QP have been disappointing. This is actually one of the reasons which motivated this work, and the use of our AL-based techniques.
>
> **One application of the QP layers is to use it as a building block for existing iterative algorithms (similar to the approach in [R1] and [R2]). The authors are suggested to analyze the convergence performance of the iterative algoritms when using the propsoed conservative Jacobian as a building block.**
>
> We thank also the reviewer for this question. We will add these two very relevant references in our related work section.
>
> The performance of the iterative algorithm using conservative Jacobian as a building block is evaluated in our experiments in Section 4.1.2, and also in our appendix (see the different "pedagogical examples" of Appendix C.1). Other works also carry out this type of convergence analysis [2,3].
>
> We would like to point out that the relatively recent notion of Conservative Jacobian (CJ) has been introduced to help explain better the behavior of Stochastic Gradient Descent-Based algorithms applied to nonsmooth functions in Deep Learning frameworks. For example, the CD3 algorithm detailed in [R1] may already use implicitly CJs. Indeed, a ReLU function is path differentiable, hence, the gradient step proposed in Section 3.2 for the inequality correction may actually be a CJ, if the functions considered in the composition (i.e., $g_x$ and $\phi_x$) are also path differentiable. This may explain the current performance of this algorithm.
>
> ## REFERENCES
>
> [1] Alice Chiche and Jean Charles Gilbert. How the augmented Lagrangian algorithm can deal with an infeasible convex quadratic optimization problem. Journal of Convex Analysis, 23(2), 2016.
>
> [2] Jérôme Bolte, Tam Le, Edouard Pauwels, and Antonio Silveti-Falls. Nonsmooth implicit differentiation for machine-learning and optimization.
>
> [3] Jérôme Bolte, Edouard Pauwels, Samuel Vaiter, Automatic differentiation of nonsmooth iterative algorithm

---

### Meta-Review · Area_Chair_RfFP · 2023-12-06

**Metareview:**

Summary: The submission presents an Augmented Lagrangian based approach to compute the gradients of the optimal solution of a convex quadartic program with respect to its inputs. Unlike previous methods, it does not assume that the program is feasible during training.

+ The problem is important and the proposed method could be generalizable to other settings not studied in the submission.
+ The empirical results are impressive.

- The paper lacks clarity.
- The significance of the theoretical results is not clear.

**Justification For Why Not Higher Score:**

- The paper lacks clarity so it might not be accessible for a wider audience.

- The significance of the theoretical results is not clear.

**Justification For Why Not Lower Score:**

+ The problem is important and the proposed method could be generalizable to other settings not studied in the submission.

+ The empirical results are impressive.

---

### Decision · Program_Chairs · 2024-01-16

Accept (spotlight)